# One-step syntheses of diaza-dioxa-fenestranes via the sequential (3 + 2) cycloadditions of linear precursors and their structural analyses

Shinichiro Fuse [1] ✉, Hiroki Ishikawa [1], Hiroshi Kitamura[1], Hisashi Masui [1] & Takashi Takahashi[2]

Fenestranes, in which four rings share one carbon atom, have garnered much attention because of their flattened quaternary carbon centers. In addition, the rigid and nonplanar structures of heteroatom-containing fenestranes are attractive scaffolds for pharmaceutical applications. We report one-step syntheses of diaza-dioxa-fenestranes via the sequential (3 + 2) cycloadditions. Our synthesis employs readily synthesizable, nonbranched acyclic allenyl precursors that facilitate sequential cycloaddition reactions. We report the synthesis of 22 heteroatom-containing and differently substituted fenestranes with rings of varying sizes. The prepared diaza-dioxa-fenestranes are subjected to X-ray crystallography and DFT calculations, which suggest that replacing the carbon atoms at the non-bridgehead positions in the fenestrane skeleton with nitrogen and oxygen atoms results in a slight flattening of the quaternary carbon center. Moreover, one of our synthesized *c,c*-[5.5.5.5]fenestranes containing two isoxazoline rings possesses the flattest quaternary carbon center among previously synthesized heteroatom-containing fenestrane versions.

Synthetic approaches for constructing complex skeletons from simple starting materials via the formation of multiple bonds in a single step are very useful in organic synthesis. Sequential cycloaddition reactions are particularly effective because they facilitate the multiple formation of bonds, both regio- and stereoselectively, in one step[1–3]. In addition, no leaving groups are required, thereby intrinsically conferring excellent atom efficiency.

One carbon atom is shared by four rings in fenestrane, which is a feature that has garnered much attention because of the resultant highly frustrated and flattened quaternary carbon centers[4–9]. How structurally modifying fenestrane influences its flattening has been investigated both experimentally and computationally[5]. Experimental evidence for such an influence, however, has been hampered by difficulties encountered during fenestrane synthesis[4,5]. A number of

biologically active natural products containing fenestrane[10,11] and heteroatom-containing fenestrane[12–18] motifs have been isolated. In addition, the rigid and nonplanar structures of heteroatom-containing fenestranes are attractive scaffolds for pharmaceutical applications. Therefore, the development of efficient synthetic approaches for (heteroatom-containing) fenestranes is an important objective.

A one-step sequential cycloaddition-based approach can effectively be used to construct (heteroatom-containing) fenestrane **A** (Fig. 1)[4]. Previously reported syntheses can be categorized into one of three approaches (Fig. 1a–c). The first approach involves the use of precursor **B**, which contains one or two rings present in the tetracyclic fenestrane structure (Fig. 1a). This approach employs sequential cycloaddition reactions with precursor **B**, resulting in the formation of fenestrane **A**. Denmark et al. reported the syntheses of two aza-dioxa-

[1]Graduate School of Pharmaceutical Sciences, Nagoya University, Nagoya, Japan. [2]Graduate School of Infection Control Sciences, Kitasato University, Tokyo, Japan. ✉e-mail: fuse.shinichiro.z3@f.mail.nagoya-u.ac.jp

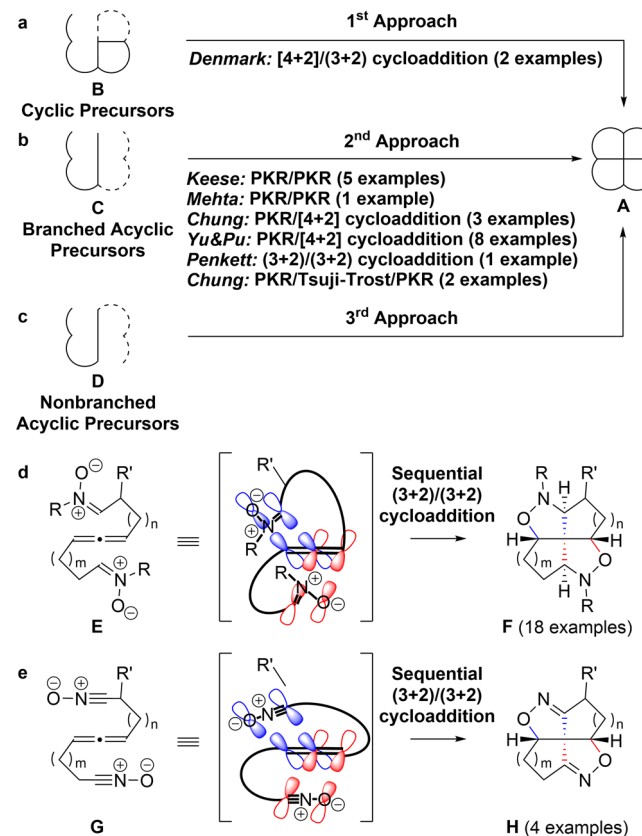

**Fig. 1 | Sequential cycloaddition-based approaches for synthesizing fenestranes. a** Sequential cycloaddition of precursor **B** containing one or two rings found in the fenestrane tetracycle (the first approach). **b** Sequential cycloaddition involving branched acyclic precursor **C** (the second approach). **c** Sequential cycloaddition of nonbranched acyclic precursors **D** (the third approach). **d** Our developed sequential (3 + 2)/(3 + 2) cycloaddition reaction involving nonbranched acyclic nitrone precursors **E** (the third approach). **e** Our developed sequential (3 + 2)/(3 + 2) cycloaddition reaction involving nonbranched acyclic nitrile oxide precursors **G** (the third approach).

fenestranes via Lewis-acid-mediated sequential [4 + 2]/(3 + 2) cycloaddition reactions based on the first approach[19]. J. Suffert et al. reported synthesis of six fenestranes via 8p-6p electrochemical cyclization from cyclic precursors that is relevant to the first approach, although this report did not use sequential cycloadditions[20]. The second approach uses branched acyclic precursor **C** that sequentially cycloadds to afford the corresponding fenestranes (Fig. 1b). Keese et al.[21,22] and Mehta et al.[23] reported the syntheses of five fenestranes[21,22] and one dioxa-fenestrane[23], respectively, via sequential Pauson–Khand reactions (PKRs) involving branched acyclic precursors, based on the second approach. Chung et al.[24] and Chen et al.[25] also reported the syntheses of three[24] and eight oxa-fenestranes[25] using similar sequential PKR/[4 + 2] cycloaddition chemistry, while Penkett et al. reported the synthesis of a dioxa-fenestrane using unique photochemical double (3 + 2) cycloaddition chemistry[26]. Chung et al. reported the elegant synthesis of two fenestranes using a PKR/Tsuji–Trost-reaction/PKR sequence starting from a branched acyclic precursor **C**[27]. Koshikawa et al. reported an elegant synthesis of eleven fenestranes via sequential (3 + 2) cycloaddition/carbenoid transfer/C-H insertion that is relevant to the second approach, although this report did not use sequential cycloadditions in a strict sense[28].

The syntheses mentioned above are plagued by issues such as low yields, a limited substrate scope, or the need for a substantial number of synthetic steps to prepare the necessary precursors. Of note, no previous synthetic protocol has successfully achieved the construction

of (heteroatom-containing) fenestranes with different ring sizes through a sequential cycloaddition approach. Although Chen et al. obtained both [5.5.5.5]oxafenestrane and [5.5.5.6]oxafenestrane as a mixture from the Rh-catalyzed sequential PKR/[4 + 2]cycloaddition of single trieneyne precursor, this was not selective reaction[25].

The third approach involves the sequential cycloaddition of the structurally simplest nonbranched acyclic precursor **D** to yield fenestrane **A** (as depicted in Fig. 1c). While this approach is appealing due to its use of easily prepared precursors, to the best of our knowledge, there are no reports documenting the utilization of this approach in previous research.

Herein, we present the inaugural one-step synthesis of differently substituted diaza-dioxa-fenestranes labeled as **F** and **H**, featuring rings of varying sizes. This synthesis is achieved using nonbranched acyclic precursors **E** and **G** through sequential (3 + 2)/(3 + 2) cycloaddition chemistry, following the principles of the third approach (Fig. 1d, e). We designed nonbranched acyclic allene precursors **E** and **G** containing nitrones and nitrile oxides, respectively, for use in sequential cycloaddition chemistry. The two orthogonal p-orbitals of an allene were anticipated to facilitate the challenging construction of the four fenestrane rings. Our approach facilitated the creation of a broad array of structurally diverse precursors. These precursors were subsequently utilized to synthesize a total of 22 diaza-dioxa-fenestranes. A structural analysis of the synthesized fenestranes indicated that the substitution of carbon atoms in the fenestrane framework with nitrogen and oxygen atoms played a role in the flattening of the quaternary carbon center. Notably, one of the synthesized diaza-dioxa-fenestranes exhibited the flattest quaternary carbon center among all previously synthesized heteroatom-containing fenestrane derivatives.

## Results

Our initial investigation focused on the one-step sequential cycloaddition reaction of nitrone-containing allene **3a** that was generated in situ from bisaldehyde **1a** (prepared from commercially available 4-pentyn-1-ol in 5 steps. See section 2.3 of the Supplementary Information for details.) and *N*-benzylhydroxylamine hydrochloride (Table 1), with solvents, including ethanol, toluene, 1,4-dioxane, and TCE, examined in the absence of a base (entries 1–3; see Supplementary Table 1 for details). The use of TCE led to the highest yield of the desired racemic [5.5.5.5]diaza-dioxa-fenestrane **2a** (entry 3, 89%). We also explored the impact of different bases, including Et₃N, *i*-Pr₂NEt, pyridine, and NaHCO₃ (entries 4–7). Remarkably, the use of Et₃N resulted in an excellent yield (91%) for **2a** (entry 4). However, when *i*-Pr₂NEt (as a stronger base) was employed, the yield of **2a** dropped significantly (19%), likely due to undesired base-mediated reactions involving bisaldehyde **1a** (entry 5). As expected, nitrone-containing allene **3a** underwent sequential cycloaddition to afford diaza-dioxa-fenestrane **2a** in high yield.

We then investigated the one-step sequential cycloaddition reaction of nitrile oxide **6a**, which was generated in situ from bisoxime **4a** (prepared from bisaldehyde **1a** in one step. See section 2.3 of the Supplementary Information for details.) (Table 2). Solvents, including CH₂Cl₂, ethanol, THF, and toluene, were examined in the presence of a 10% aqueous solution of NaOCl and Et₃N. The use of CH₂Cl₂ resulted in a good yield of the desired racemic double-bond-containing [5.5.5.5] diaza-dioxa-fenestrane **5a** (entry 1, 69%). The use of ethanol did not afford any of the desired product (entry 2), despite the high solubility of NaOCl in this solvent. The desired product **5a** was obtained, in yields of 15% and 20% when THF and toluene was used, respectively (entries 3 and 4). Bases, including Et₃N, pyridine, NaHCO₃, and *i*-Pr₂NEt were examined using CH₂Cl₂ as the solvent (entries 1 and 5–7); once again the use of Et₃N afforded the highest yield (entry 1, 69%). The use of *i*-Pr₂NEt afforded **5a** in acceptable yield (entry 7, 59%) during the cycloaddition of bisnitrile oxide **6a**, in contrast to the yield obtained using nitrone **3a**. The desired product **5a** was also obtained in an

**Table 1 | Investigating the one-step sequential (3 + 2)/(3 + 2) cycloaddition chemistry of a nitrone-containing allene**

| Entry | Base | Solvent | Temp. [°C] | Yield[a] [%] |
|---|---|---|---|---|
| 1 | None | Toluene | 110 | 57 |
| 2 | None | 1,4-dioxane | 100 | 75 |
| 3 | None | TCE | 110 | 89 |
| 4 | Et$_3$N | TCE | 110 | 91 |
| 5 | i-Pr$_2$NEt | TCE | 110 | 19 |
| 6 | Pyridine | TCE | 110 | 73 |
| 7 | NaHCO$_3$ | TCE | 110 | 77 |

TCE 1,1,2-trichloroethane.
[a]Isolated yield.

acceptable yield in the absence of the base (entry 8, 62%). At this point, we also examined the effect of temperature (0–100 °C) using haloalkane solvents (entries 9–13); the use of DCE at 80 °C afforded the highest yield (entry 12, 72%). Accordingly, we successfully developed a sequential cycloaddition-based approach using two types of allenes **3a** and **6a** containing nitrone and nitrile oxide, respectively. The structures of **2a** and **5a** were unambiguously determined by X-ray crystallography (see section 11 of the Supplementary Information for details); structural-analysis details are discussed below.

The substrate scope of the developed one-step sequential cycloaddition chemistry involving nitrones **3** was subsequently examined (Fig. 2a). While *N*-alkyl substituted fenestranes **2b**–**2d** were obtained in good yields (64–76%), only a trace amount of *N*-*t*-butyl-substituted fenestrane **2e** was obtained, while *N*-4-methoxybenzyl-substituted fenestrane **2f** was obtained in moderate yield (48%). We next examined the functional-group tolerance of this reaction. Good yields (68–71%) were obtained in the syntheses of **2g** containing Ph−Br bonds, **2h** containing O−Si bonds, and **2i** and **2j** containing acid-labile THP and furyl groups, respectively. Fenestranes **2k** and **2l** containing either acid-labile Boc-carbamate or *t*-Bu-ester moieties, and either base-labile methyl ester or Fmoc-carbamate moieties, were also obtained in good yields (73 and 62%). On the other hand, *N*-phenyl substituted fenestrane **2m** was only obtained in moderate yield (35%), while *C*,*N*-alkyl(oxy)-substituted fenestranes **2n**–**2q** were obtained in acceptable-to-good yields (56–70%) as mixtures of diastereomers. Although the diastereomers of **2o** were inseparable, **2n, 2p,** and **2q** were readily separated into their diastereomers by silica-gel column chromatography. The stereochemistry of each diastereomer was determined via ¹H NMR, COSY, and NOESY spectroscopy, along with DFT calculations (see section 8 of the Supplementary Information for details). [5.5.5.6]Diaza-dioxa-fenestrane **2r** and [5.6.5.6]diaza-dioxa-fenestrane **2s** were obtained in yields of 65 and 12%, respectively, while [5.7.5.7]diaza-dioxa-fenestrane **2t** was not obtained.

We next examined the substrate scope of the one-step sequential cycloaddition chemistry involving nitrile oxides **6** (Fig. 2b). *C*-Alkyl(oxy)-substituted fenestranes **5b** and **5c** were obtained in good yields (69 and 66%) as mixtures of diastereomers. While **5b** was unable to be separated nor was its diastereomeric ratio able to be determined, **5c** was readily separated into its diastereomers via silica-gel column chromatography. The stereochemistry of each diastereomer was determined by ¹H NMR, COSY, and NOESY spectroscopy, along with DFT calculations (see section 9 of the Supplementary Information for details). To our delight, [5.5.5.6]diaza-dioxa-fenestrane **5d** was obtained in moderate yield (36%), whereas [5.6.5.6]diaza-dioxa-fenestrane **5e** was not obtained. While one-step sequential cycloaddition chemistry involving nitrones **3** enabled the synthesis of diaza-dioxa-fenestranes containing up to two six-membered rings, the developed chemistry involving nitrile oxides **6** enabled the synthesis of diaza-dioxa-fenestranes containing only one six-membered ring. The latter chemistry appeared to be more significantly affected by the ring size. Using the developed approach, fenestranes with different ring sizes were constructed through sequential cycloaddition.

The prepared THP-, TBDPS-, Boc-, Fmoc-, *t*-Bu-, and Bn-protected diaza-dioxa-fenestranes can be readily derivatized via deprotection and subsequent chemical modification. In addition, the aryl-Br bond in fenestrane **2g** can be directly activated in the presence of transition-metal catalysts for further derivatization. Accordingly, **2g** was subjected to Suzuki−Miyaura, Sonogashira−Hagihara, and Mizoroki−Heck coupling, which afforded the desired products **7a**–**7d** in acceptable-to-excellent yields (Fig. 2c, 57–96%). Moreover, the reactive N−O and C=N bonds in the diaza-dioxa-fenestranes were further derivatized; reductive cleavage of the N−O bond in isoxazolidine **2a** afforded spirobicycle **8** in excellent yield (Fig. 2d, 93%). Spiro[4.4]nonane **8**, which was densely functionalized by two amino groups and two hydroxy groups at the neopentyl positions, was obtained as a single diastereomer. Mono- and

**Table 2 | Investigating the one-step sequential (3 + 2)/(3 + 2) cycloaddition chemistry of a nitrile oxide**

| Entry | Base | Solvent | Temp. [°C] | Yield[a] [%] |
|---|---|---|---|---|
| 1 | Et$_3$N | CH$_2$Cl$_2$ | r.t. | 69 |
| 2 | Et$_3$N | Ethanol | r.t. | n.d.[c] |
| 3 | Et$_3$N | THF | r.t. | 15 |
| 4 | Et$_3$N | Toluene | r.t. | 20 |
| 5 | Pyridine | CH$_2$Cl$_2$ | r.t. | 50 |
| 6 | NaHCO$_3$ | CH$_2$Cl$_2$/H$_2$O[b] | r.t. | 12 |
| 7 | $i$-Pr$_2$NEt | CH$_2$Cl$_2$ | r.t. | 59 |
| 8 | None | CH$_2$Cl$_2$ | r.t. | 62 |
| 9 | Et$_3$N | CH$_2$Cl$_2$ | 0 | n.d.[c,d] |
| 10 | Et$_3$N | CH$_2$Cl$_2$ | 40 | 66 |
| 11 | Et$_3$N | DCE | 60 | 63 |
| 12 | Et$_3$N | DCE | 80 | 72 |
| 13 | Et$_3$N | TCE | 100 | 66 |

*THF* tetrahydrofuran, *DCE* 1,2-dichloroethane, *r.t.* room temperature.
[a]Isolated yield.
[b]5:1 CH$_2$Cl$_2$:H$_2$O was used to dissolve NaHCO$_3$.
[c]**5a** was not detected.
[d]A monocyclized product was detected.

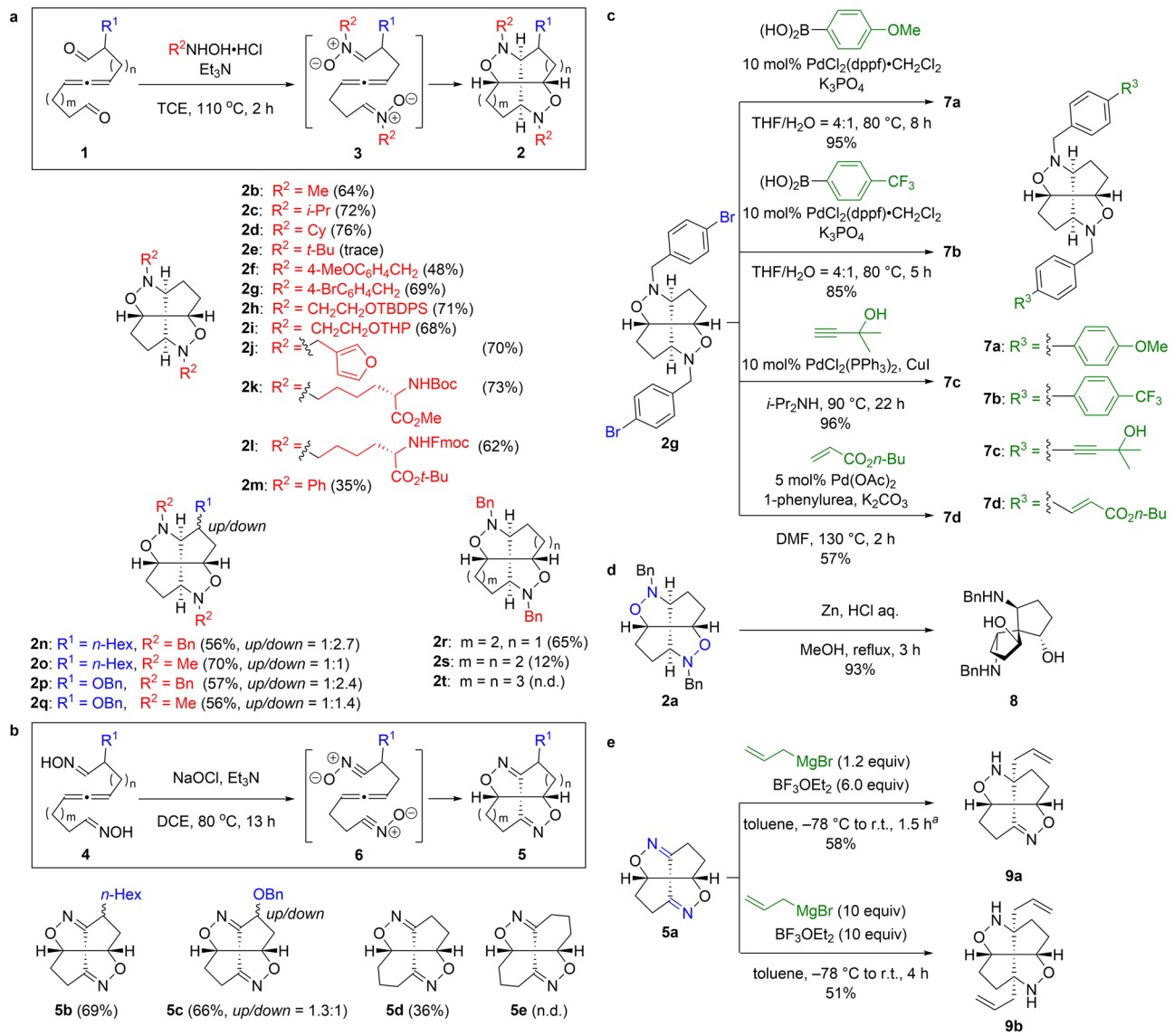

**Fig. 2 | Substrate scope.** Substrate scope of our developed one-step sequential cycloaddition approach from **a** nitrone **3** and **b** nitrile oxide **6**. **c** Derivatizing **2g** using Pd-catalyzed cross-coupling reactions. **d** Reductive cleavage of the N−O bond in isoxazolidine **2a**. **e** Mono- and bis-allylation of isoxazoline **5a** using a Grignard reagent. [a]Addition of reagents at −78 °C and warmed to r.t. was repeated 5 times. Cy = cyclohexyl; TBDPS = t-butyldiphenylsilyl; THP = tetrahydropyranyl; Boc = t-butyloxy carbonyl; Fmoc = 9-fluorenylmethoxycarbonyl; Bn = benzyl; dppf = (diphenylphosphino)ferrocene; DMF = N,N-dimethylformamide.

bis-allylated isoxazolidines **9a** and **9b** were selectively obtained by 1,2-addition using different amounts of an allyl Grignard reagent to iso-xazoline **5a**; both **9a** and **9b** were obtained diastereoselectively in acceptable yields (Fig. 2e, 58 and 51% yield, respectively). These results clearly demonstrate that our approach facilitates the creation of structurally diverse and complex heterocyclic compounds.

As previously discussed, the flattened fenestrane quaternary carbon center, which is shared by four rings, has garnered much attention. The extent of flattening of such a quaternary carbon center can be evaluated from its two opposing angles (α and β in Fig. 3)[4,5], these angles increase as the quaternary carbon center flattens. Keese et al. reported α and β values of 116.2°[29] and 113.8°[5] for c,c,c,c-[5.5.5.5]fenestrane (**10**) based on electron diffractometry and semi-empirical calculations, respectively (Fig. 3a). The flattest reported fenestranes **11** and **12** have significantly larger angles (Fig. 3b, α = 134.9°, β = 119.2°[30]; Fig. 3c, α = 129.2°, β = 128.3°[31]) than non-distorted quaternary carbon (109.5°, Fig. 3d).

The racemic [5.5.5.5]- and [5.5.5.6]fenestranes **2a** and **2r**, respectively, containing isoxazolidine rings and the [5.5.5.5]fenestrane **5a** containing isoxazoline rings were analyzed using X-ray

crystallography, which revealed α and β values consistent with those of the most stable conformers determined by DFT at the B3LYP[32]/6-31 G + (d,p)[33–36] level of theory (Fig. 3e–g). A comparison of the quaternary-carbon angles in fenestrane **10** with those in diaza-dioxa-fenestrane **2a** (Fig. 3a, e) reveals that replacing the carbon atoms at the non-bridgehead positions in the fenestrane skeleton with nitrogen and oxygen atoms results in slight flattening of the quaternary carbon center. A comparison of the angles in [5.5.5.5]diaza-dioxa-fenestrane **2a** with those in [5.5.5.6]diaza-dioxa-fenestrane **2r** (Fig. 3e, f) reveals that ring expansion reduces the degree of flattening of the quaternary carbon center, which is consistent with the previously reported tendency[5]. In addition, a comparison of the angles in [5.5.5.5]fenes-trane **2a** containing isoxazolidine rings with those in [5.5.5.5]fenestrane **5a** containing isoxazoline rings (Fig. 3g) reveals that the introduction of double bonds at the bridgehead positions results in flattening of the quaternary carbon center, which is also consistent with the previously reported tendency[5]. However, the observed angles in **5a** (Fig. 3g, α = 134.7°, β = 114.9°) are very large and comparable to those of fenestrane **11** and **12**, which are among the flattest fenestranes known

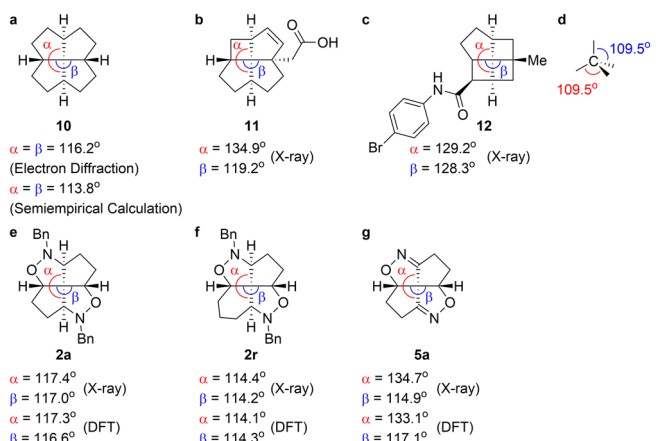

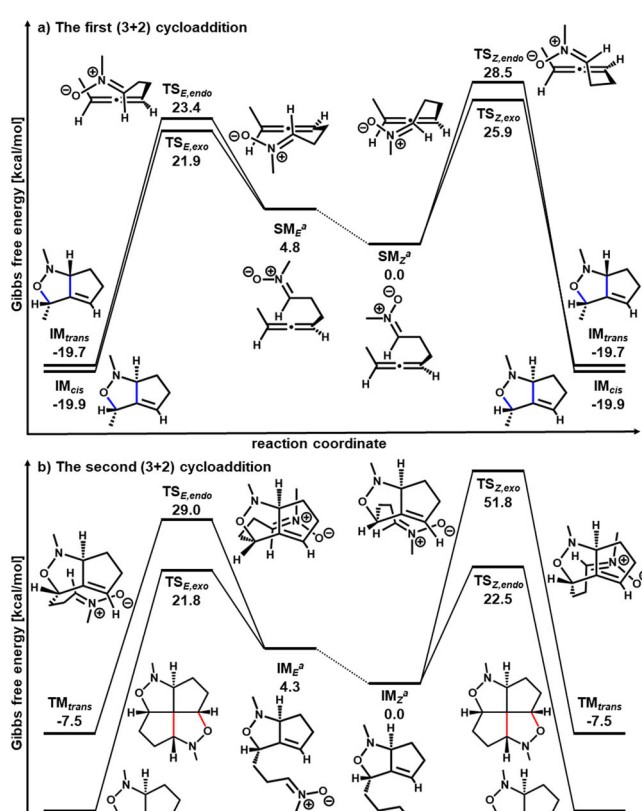

**Fig. 3 | Fenestrane structures.** Opposing angles: **a** in *c,c,c,c*-[5.5.5.5]fenestrane **10**, **b**, **c** in the flattest previously synthesized fenestrane **11** and **12**, and **d** for a non-distorted quaternary carbon center. **e–g** Experimentally determined and calculated opposing angles in *c,c,c,c*-[5.5.5.5]fenestrane **2a** containing isoxazolidine rings, *c,c,c,c*-[5.5.5.6]fenestrane **2r** containing isoxazolidine rings, and *c,c*-[5.5.5.5]fenestrane **5a** containing isoxazoline rings.

(Fig. 3b, c). Compound **5a** contains the flattest quaternary carbon center among heteroatom-containing fenestranes discovered thus far.

We performed a conformation search for fenestranes **2b** and **5a**. To reduce the calculation cost, fenestrane **2b** with methyl groups was used instead of **2a** with benzyl groups. The four most stable conformers 1–4 of **2b** and the two most stable conformers 1 and 2 of **5a** are shown with relative energy levels and α and β values in Supplementary Table 9 of the Supplementary Information. The chemical structure of **2a** experimentally observed via X-ray crystallographic analysis (α = 117.4°, β = 117.0°) was consistent with the calculated most stable conformer 1 of **2b** (α = 117.6°, β = 116.8°). Although the conformers 2–4 of **2b** with more flattened quaternary carbon centers were found in the conformation search (Supplementary Table 9), they were less stable. Only two conformers with almost consistent α and β values and similar structures were found in the case of **5a**. These results indicated that the compound **5a** has a very rigid structure.

## Discussion

In this study, all-*cis*-fused diastereomers (*c,c,c,c*-[5.5.5.5]fenestrane **2** containing isoxazolidine rings and *c,c*-[5.5.5.5]fenestrane **5** containing isoxazoline rings) were obtained exclusively. No generation of *trans*-fused fenestranes was observed in the NMR analysis of crude products, for the syntheses of both **2** and **5**. DFT calculation at the B3LYP[36]/6-31 G + (d,p)[33–36] level of theory was performed to elucidate the mechanism of the sequential (3 + 2) cycloadditions on the basis of the report of Nguyen et al.[37]. Alkyl substituents were replaced with methyl groups to reduce the calculation cost.

We performed a DFT calculation of the sequential cycloaddition affording **2** (Fig. 4). The calculation results for the first (3 + 2) cycloaddition of the nitrone precursors with the *E*-configuration (SM$_E$) and *Z*-configuration (SM$_Z$) affording IM$_{cis}$ (intermediate for the all-*cis*-fused diastereomer of the fenestrane) and IM$_{trans}$ (intermediate for the *trans*-fused diastereomer of the fenestrane) are shown in Fig. 4a. The comparison of four pathways (*exo*- and *endo*-cyclizations of SM$_E$ and SM$_Z$ substrates) suggested that the *exo*-cyclization of SM$_E$ via the transition state TS$_{E,exo}$ is the most energetically favored pathway affording IM$_{cis}$. The suggested most energetically favored pathway affording IM$_{trans}$ is *endo*-cyclization of SM$_E$ via the transition state TS$_{E,endo}$. The calculated activation energy difference is 1.5 kcal/mol, and the TS$_{E,exo}$ leading to IM$_{cis}$ is energetically favored over TS$_{E,endo}$ leading to IM$_{trans}$. As expected, the two orthogonal p-orbitals of the

**Fig. 4 | DFT calculation for (3 + 2) cycloadditions. a** DFT calculation for the first (3 + 2) cycloaddition. **SM$_E$**: nitrone substrate with the *E*-configuration, **SM$_Z$**: nitrone substrate with the *Z*-configuration, **TS$_{E,exo}$**: transition state of *exo*-cyclization of **SM$_E$**, **TS$_{E,endo}$**: transition state of *endo*-cyclization of **SM$_E$**, **TS$_{Z,exo}$**: transition state of *exo*-cyclization of **SM$_Z$**, **TS$_{Z,endo}$**: transition state of *endo*-cyclization of **SM$_Z$**. **IM$_{cis}$**: bicyclic intermediate for the all-*cis*-fused diastereomer of the fenestrane. **IM$_{trans}$**: bicyclic intermediate for the *trans*-fused diastereomer of the fenestrane. **b** DFT calculation for the second (3 + 2) cycloaddition. **IM$_E$**: nitrone intermediate with the *E*-configuration, **IM$_Z$**: nitrone intermediate with the *Z*-configuration, **TS$_{E,exo}$**: transition state of *exo*-cyclization of **IM$_E$**, **TS$_{E,endo}$**: transition state of *endo*-cyclization of **IM$_E$**, **TS$_{Z,exo}$**: transition state of *exo*-cyclization of **IM$_Z$**, **TS$_{Z,endo}$**: transition state of *endo*-cyclization of **IM$_Z$**. **TM$_{trans}$**: *trans*-fused diastereomer of the fenestrane. **TM$_{cis}$**: all-*cis*-fused diastereomer of the fenestrane. ªNeither the energy barrier for tautomerizations between **SM$_E$** and **SM$_Z$** nor **IM$_E$** and **IM$_Z$** were calculated.

allene appear to facilitate the approach of reaction sites in TS$_{E,exo}$. IM$_{cis}$ appears to have a similar energy level to IM$_{trans}$.

The calculation results for the second (3 + 2) cycloaddition of the nitrone intermediates with the *E*-configuration (IM$_E$) and *Z*-configuration (IM$_Z$) affording TM$_{cis}$ (experimentally obtained all-*cis*-fused diastereomer) and TM$_{trans}$ (experimentally not obtained *trans*-fused diastereomer) are shown in Fig. 4b. The comparison of four pathways (*exo*- and *endo*-cyclizations of IM$_E$ and IM$_Z$ intermediates) again suggests that the *exo*-cyclization of IM$_E$ via the transition state TS$_{E,exo}$ is the most energetically favored pathway affording TM$_{cis}$. In addition, the *endo*-cyclization of IM$_Z$ via the transition state TS$_{Z,endo}$ was also suggested as the energetically plausible pathway in the case of second cycloaddition because the calculated activation energy difference between TS$_{E,exo}$ and TS$_{Z,endo}$ was 0.7 kcal/mol. The most energetically favored pathway affording TM$_{trans}$ was again suggested to be *endo*-cyclization of IM$_E$ via the transition state TS$_{E,endo}$. The calculated activation energy difference between TS$_{E,exo}$ and TS$_{E,endo}$ was 7.2 kcal/mol, and the TS$_{E,exo}$ affording TM$_{cis}$ was energetically favored over TS$_{E,endo}$ affording TM$_{trans}$. Moreover, **TM$_{cis}$** was significantly more

stable than **TM**_*trans*_. These results explain the reason for obtaining only the all-*cis* diastereomer **TM**_*cis*_.

We could not calculate transition states affording a *trans*-fused diastereomer in the case of (3 + 2) cycloaddition of the nitrile oxide precursor, because of the highly strained structure. The calculated pathway affording the all-*cis*-fused diastereomer of the fenestrane **5a** is presented in Supplementary Table 10 in the Supplementary Information.

We developed a one-step sequential (3 + 2) cycloaddition approach for the synthesis of diaza-dioxa-fenestranes that uses structurally simple, readily synthesizable, nonbranched acyclic allenyl precursors that facilitate sequential cycloaddition reactions. Twenty-two structurally diverse, heteroatom-containing, and variously substituted fenestranes, **2** and **5**, with rings of different sizes, were successfully prepared. In addition, **2a, 2g**, and **5a** were further structurally modified to afford more-functionalized derivatives **7a**–**7d**, and **9a**–**9b**. Spiro[4.4]nonane **8**, which was densely functionalized by two amino groups and two hydroxy groups at the neopentyl positions, was obtained as a single diastereomer from reductive cleavage of N-O bonds of **2a**. The prepared diaza-dioxa-fenestranes **2a, 2r**, and **5a** were analyzed by X-ray crystallography and DFT calculations. Experimentally determined angles α and β were found to be consistent with those calculated using DFT. Our results indicate that replacing the carbon atoms at the non-bridgehead positions in the fenestrane skeleton with nitrogen and oxygen atoms slightly flattens the quaternary carbon center. In addition, we experimentally confirmed that ring expansion reduces the degree of flattening, whereas the introduction of double bonds at the bridgehead positions of a fenestrane increases the degree of flattening. Moreover, the synthesized *c,c*-[5.5.5.5]fenestrane **5a** containing isoxazoline rings exhibited the flattest quaternary carbon center among previously synthesized heteroatom-containing fenestrane versions. This synthetic approach is expected to drive the development of structurally diverse and unique heteroatom-containing fenestranes, and the observed effects of chemical modification on the flattening of the quaternary carbon centers are expected to contribute to our further understanding of frustrated and flattened carbon centers.

## Methods

### General procedure for cycloaddition via nitrone

**Method A**: Et₃N (86.6 μL, 0.625 mmol, 2.50 equiv.) and hydroxylamine hydrochloride (0.625 mmol, 2.50 equiv.) were added to a stirred solution of allene bisaldehyde **1** (0.250 mmol, 1.00 equiv.) in TCE (50.0 mL) at room temperature under argon. The mixture was stirred at 110 °C for 2 h, cooled to room temperature, and then quenched with water. The aqueous layer was extracted with $CH_2Cl_2$ (3×) and the combined organic layers were washed with brine, dried over $Na_2SO_4$, filtered, and the filtrate was concentrated under reduced pressure. The residue was purified by silica-gel column chromatography or preparative TLC to give the corresponding fenestrane **2**.

**Method B**: Hydroxylamine (0.625 mmol, 2.50 equiv.) was added to a stirred solution of allene bisaldehyde **1** (0.250 mmol, 1.00 equiv.) in TCE (50.0 mL) at room temperature under argon. The mixture was stirred at 110 °C for 2 h, cooled to room temperature, and concentrated under reduced pressure. The residue was purified by silica-gel column chromatography to afford fenestrane **2**.

### General procedure for cycloaddition reactions involving nitrile oxides

Aqueous NaOCl (12 wt% 1.55 mL, 2.50 mmol, 10.0 equiv.) and Et₃N (347 μL, 2.50 mmol, 10.0 equiv.) were added to a stirred solution of bisoxime **4** (0.250 mmol, 1.00 equiv.) in DCE (50.0 mL) at room temperature under argon. The mixture was stirred at 80 °C for 13 h, cooled to room temperature, and subsequently diluted with water. The aqueous layer was extracted with $CH_2Cl_2$ (3×), and the combined organic layers were washed with brine, dried over $Na_2SO_4$, filtered, and the filtrate was concentrated under reduced pressure. The residue was purified by using silica-gel column chromatography to afford fenestrane **5**.

## Data availability

The authors declare that the data for this study are available within the manuscript and its Supplementary Information files and are also available from the corresponding author. The nuclear magnetic resonance (NMR) spectra, experimental procedures, characterization results for all products, and DFT calculations are presented in the Supplementary Information file. Source data in DFT calculations are provided with this paper. The X-ray crystallographic coordinates for the structures reported in this article have been deposited at the Cambridge Crystallographic Data Centre (CCDC), under deposition numbers CCDC-2233428 (for **2a**), CCDC-2233429 (for **2r**), and CCDC-2233425 (for **5a**). These data can be obtained free of charge from the Cambridge Crystallographic Data Centre via www.ccdc.cam.ac.uk/structures. Source data are provided with this paper.

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

## Acknowledgements
We thank Dr. Takeshi Yasui (Department of Basic Medicinal Sciences, Graduate School of Pharmaceutical Sciences, Nagoya University) for technical X-ray crystallography assistance. Pd/C was generously pro-vided by Kawaken Fine Chemicals, Japan. This work was partially sup-ported by JSPS Fellows (23KJ1102, H.K.), and the Research Support Project for Life Science and Drug Discovery [Basis for Supporting Inno-vative Drug Discovery and Life Science Research (BINDS)] from the Japan Agency for Medical Research and Development (AMED) under Grant Number JP23ama121044 (S.F.).

## Author contributions
T.T. conceived of this study. H.M. performed the initial experimental study. H.I. performed most of all the experimental studies and DFT cal-culation of fenestranes. H.K. performed DFT calculation of the sequential cycloaddition and analyzed the reaction pathway. H.M. and H.I. per-formed the X-ray crystallographic analysis. H.M. and S.F. supervised the conduct of this study. S.F. drafted the original manuscript. All authors have reviewed and approved the final version of the manuscript.

## Competing interests
The authors declare no competing interests.
