## [Peer Review File · Nature Communications]

One-step syntheses of diaza-dioxa-fenestranes via the sequential (3+2) cycloadditions of linear precursors and their structural analysesREVIEWER COMMENTS

Reviewer #1 (Remarks to the Author):

please see file attached

Reviewer #2 (Remarks to the Author):

The authors report in great detail and in a remarkably broad scope on the synthesis of heterocyclic fenestranes of unprecedented constitution. The work presented in this manuscript is sound, of high scientific and technical quality. Also, it has a very high level of originality and deals with a particularly fascinating topic of organic chemistry.

The molecular skeleton of fenestranes is characterized by a tetracyclic core bearing a central atom shared by all of the four rings. The fenestrane skeleton can be regarded as comprising two crosswise fused spirane units or, alternatively, two laterally fused bicyclo[x.y.0]alkane units. The stereochemistry of fusion of the four rings can vary, but most of the known fenestranes have all-cis-fused configuration (*c,c,c,c*), which is energetically most favorable because of by far the least steric strain. In a rather small subset of fenestranes that are known by experiment, one or even two of the four peripheral bridgeheads are unsaturated, which considerably increases the strain. However, fenestrane chemistry has gained most attention because of the flattening effect of the quadruple ring fusion on the configuration of the central carbon atom. Although true “planarization” of a tetracoordinate C-atom has never been achieved experimentally in an organic compound, the potential to construct novel fenestranes with increasing degree of flattening continues to be scientifically tempting and challenging.

Based on these considerations, the present work is highly encouraging for at least three reasons.

(1) The synthesis strategy is novel and particularly impressive, since the tetracyclic framework of the title compounds is constructed in a single step from linear precursors. In other words, none of the four rings of the target fenestrane needs to be preformed and stepwise ring assembly can be avoided. To the best of my knowledge, this approach has never been pursued before. This important progress has become reality now by employing linear allenes that bear two in-situ-generated 1,3-dipolar functionalities at their α - and ω -positions. Hence, two sequential [3 + 2]-cycloaddition reactions take place in a single synthesis set and with high efficiency. The allene-based bis-dipoles were generated from suitable allene- α,ω -dialdehydes or allene- α,ω -bis-aldoximes in, admittedly, quite elaborate stepwise syntheses.

(2) The simplicity of this crucial step allows a large chemical variability in accessing novel *N,O*-heterocyclic fenestranes. Specifically, two N-O bonds form the rims of one of the two spirocyclic subunits which renders framework chiral. One single but also more substituents can be easily incorporated owing to the fact that the central allene unit of the fenestrane precursors is constructed in a nonsymmetric manner. The use of two related but different strategies, namely, use of nitrones, $R-CH=N^+(O^-)-R'$, and nitrile oxides, $R-C\equiv N^+-O^-$, as 1,3-dipolar cycloaddition components, allows the construction of both N-saturated and N-unsaturated fenestranes. In the latter case, two C-N double bonds can be generated with ease at the opposite bridgehead positions of a strained *cis,cis*-[5.5.5.5]fenestradiene

skeleton. Further, these double bonds have been used successfully to introduce two allyl groups at the unsaturated bridgeheads. Another remarkable success – which may be overlooked easily in the given context – is the successful synthesis of a (carbocyclic!) spiro[4.4]nonane bearing one singly-bonded functional (amino and hydroxy) group at each of the four α -positions of its neopentane core (viz., compound **8** from fenestrane **2a**, Figure 2). Please see remark (9) below.

(3) The results reported in this work enables further insights into the effect of the fenestrane periphery on the flattening of configuration of the central carbon atom. As mentioned above, incorporation of heteroatoms into the fenestrane framework represents a valuable extension of this theme. Here, the elucidation of the bond angles at the central carbon atom of compounds **2a**, **2r** and **5a** represents an important progress. In particular, this hold true for fenestradiene **5a** containing two unsaturated bridgeheads. The central bonds angles found by single crystal X-ray analysis (and DFT calculations) of this structure ($\alpha = 134.7^\circ$ and $\beta = 114.9^\circ$) are comparably large and of similar size as those of a [4.5.5.5]fenestrene derivative reported by Keese et al. Moreover, the present shows that ring-size effects in the new diazadioxafenestranes are similar to those known from previous work in the carbocyclic series.

On the basis of all these considerations, I am convinced that the results presented in the manuscript add tremendously to the knowledge of fenestrane chemistry and provide inspiring implications to other, more general fields of chemistry. Nevertheless, I found a number of aspects that may deserve correction or improvement prior to acceptance for publication. These points are listed below. In particular, please note the important critique given in (8).

* * * * *

(1) Some parts of the text are somewhat inelegant and deserve improvement. Other minor items to be corrected are listed here as well. (i) Page 1 (Abstract) and elsewhere (e.g. page 5 and page 9, Discussion): The term “acyclic allenyl precursors with orthogonal p-orbitals” should be shortened because the reader will know that allene have two mutually orthogonal π -orbitals. – (ii) Page 1 (Abstract) and elsewhere: The word “planarization” should be used with care since the configuration of the central C-atom of fenestranes has remained very far from being planar. The term “flattening” would be more appropriate in most cases. – (iii) Page 1 (Abstract) and elsewhere (e.g. page 8, line 5): The chemical name “*c,c,c,c*-[5.5.5.5]isoxazolines” is incorrect (and meaningless). What is actually meant is here is “*cis,cis,cis,cis*-[5.5.5.5]festrans containing two isooxazoline units”. Please do not mix different terms of nomenclature. – (iv) Page 2, line 2: The phrase “... formation of multiple bonds in single steps ...” is incorrect and misleading. – (v) Page 2, 2nd paragraph: As to precursor **B** in Figure 1, the authors mention one single ring but the structure contains a second (dashed) one. This should be clarified in the text. Improvement along these lines is also necessary in the legend to Figure 1. – (vi) Page 6, line 6 (and maybe elsewhere, e.g. page 7, legend to Figure 2): The term “arylbromide” for the *para*-bromophenyl residue in compound **2g** is unfortunate. – (vii) Page 6, 2nd-last line of 1st paragraph: The word “although” should be replace by “while” or “whereas”. – (viii) Page 7, line 2: The phrase “... unique cyclic compounds” appears to be too vague an expression here (maybe better: “complex heteropolycyclic?”). – (ix) Pages 7-8: Just as an example, the terminology should be checked thoroughly throughout. Writing “... as the tetrahedral carbon flattens ...” is really not good.

(2) Figure 1: The novel strategy is displayed in part d of this figure but not sufficiently clearly. While the dashed bonds in **E** and **F** are obviously meant to comprise both the nitron and nitrile oxide variants, and their meaning in **F** can be understood, the presence of the two residues "R" in **E** excludes the intended perception of nitrile oxide groups. In my view, the two variants could easily be presented in two separate lines here. Also, this would favorably highlight these two variants early in the paper.

(3) Page 5 (top): It appears that a comment on entry 8 (no base) is missing.

(4) Page 5 (top): The phrase "... the higher stability of the less-acidic bisoxime **4a** ..." is misleading. I assume that the authors intend to address the lower C-H acidity of the bisoximes? Otherwise, they should exert some O-H acidity.

(5) The scheme at the top of Table 2 displays the nitrile oxide groups in a nonlinear form. In my view, linear presentation ($R-C\equiv N^+-O^-$) should be preferred.

(6) Page 6, end of 1st paragraph and/or possibly also end of 2nd paragraph: A comment on the (limited) ring-size tolerance of the synthesis methodology should be included here.

(7) Page 7, Figure 2, and Supporting Information (Experimental): While most of the data presented here are in line with those given in the SI, there seems to be a mistake in the diastereomer ration of compound **2q**. The ratio of the "up"- and "down" isomers is given as 1.4 : 1 in Figure 2 but in the Experimental, the ratio **2q** : **2q'** = 23 : 33 ($\approx 1 : 1.4$) is given. Please check.

(8) Page 8, 2nd paragraph: I do not completely agree with the discussion on the flattening effects on the conformation of the central C-atom observed by X-ray diffraction (and supported by the DFT calculations). It should be improved. Taken strictly, there are three points: (i) The statement "A comparison of the quaternary-carbon angles in fenestrane **10** with those in diaza-dioxa-fenestrane **2a** (Fig. 3, **a** and **d**) reveals that replacing the carbon atoms at the non-bridgehead positions in the fenestrane skeleton with nitrogen and oxygen atoms results in planarization of the quaternary carbon, and represents the first example of planarization through the introduction of heteroatoms." is incorrect or, at least, exaggerated. In fact, the flattening effect in **2a** is very similar to that found in the carbocyclic fenestrane **10**. The slight increase of the angles found for **2a** may be real but, even then, it may be due to packing effects. Therefore, I recommend choosing a more cautious statement. – (ii) The next statement; "A comparison of the angles in [5.5.5.5]isoxazolidine **2a** with those in [5.5.5.6]isoxazolidine **2r** (Fig. 3, **e**) reveals that ring contraction planarizes the quaternary carbon, which is consistent with the previously reported tendency.⁵" is misleading. Probably, the authors want to state that contracting the ring size from six to five (i.e., **2r** → **2a**) increases the flattening. However, five-membered rings in fenestranes may be considered a kind of standard; therefore, the statement should be reversed to express simply that incorporation of a six-membered ring in **2r** decreases the flattening. – (iii) The discussion on the "flattest" quaternary carbon atom is somewhat unsatisfying. Not only Keese's fenestrene **11** (central bond angles $\alpha = 134.9^\circ$ and $\beta = 119.2^\circ$) should be considered but also Agosta's [4.4.4.5]fenestrane ($\alpha = 129.2^\circ$ and $\beta = 128.3^\circ$, *J. Am. Chem. Soc.* **1985**, *107*, 5732). To avoid, in the same time, the slightly unclear statement "... hence, **5a** contains the flattest quaternary carbon atom among previously synthesized heteroatom-containing fenestrane versions", I suggest to replace the last sentence of this paragraph by: "However, the observed angles in **5a** (Fig. 3, **f**, $\alpha = 134.7^\circ$, $\beta = 114.9^\circ$) are very large and comparable to those of fenestrane **11**, which is among the flattest fenestranes known (Fig. 3, **b**).

Compound **5a** contains the flattest quaternary carbon atom among heteroatom-containing fenestranes known so far." Of course, displaying Agosta's [4.4.4.5]fenestrane in Figure 3 and including this detail into the discussion would be even better.

(9) Page 9, Discussion section: The "Discussion" presented here does not really discuss but, rather, conclude the results but may be acceptable. In any case, however, the "unique spirocycle **8**" does deserve some discussions, as stated above.

(10) A further remark: Compound **9b** appears to be a promising candidate for a ring-closing metathesis reaction between the two *syn*-oriented allyl groups. This would produce an interesting, novel centropentacyclic structure.

(11) And one more - at the very end: The authors give a useful introduction on the chemistry of alicyclic fenestranes and general aspects of flattening. Unfortunately, they do not mention at all that benzannulated fenestranes have also been studied in great detail and also with respect to flattening of the geometry of the central carbon atom. This aspect may be considered.

(12) Supporting Information: The supporting information is very well-presented and provides complete information on the impressive experiments and measurements. However, some details may deserve improvement of correction, still:

(a) Although the synthesis steps are given completely, a general overview in a scheme at the beginning of the SI (page S-2) would be helpful.

(b) Page S-3 (top): Numerous incorrect statements appear in the experimental procedures, such as: "After being stirred at room temperature for 2 h, the reaction was quenched ...". Referring the stirring to the reaction is not correct and not good style, given the very frequent cases of this sort throughout.

(c) Page S-4 and throughout the experimental procedures: There is no doubt that the calculated accurate mass values of the $[M + Na]^+$ ions measured in most cases are all correct. However, the statement in the form "HRMS (ESI): calcd. for $C_{13}H_{22}O_2$ ($[M+Na]^+$): 233.1512, found: 233.1514" is not correct because the calculated and measured values refer to the sodiated molecular adduct ions. Therefore, it should better read: "HRMS (ESI) of $[M + Na]^+$: calcd for $C_{13}H_{22}O_2Na^+$ 233.1512, found 233.1514" here, and the like elsewhere.

(d) Pages S-32: Similarly, the accurate mass values of the $[M + H]^+$ ions refer to the formula $C_{11}H_{19}N_2O_2^+$, not $C_{11}H_{18}N_2O_2^+$. Therefore, it should better read here: "HRMS (ESI) of $[M + H]^+$: calcd for $C_{11}H_{19}N_2O_2^+$: 441.2384, found 441.2382". - Similar corrections are needed on Page S-36 (calcd for $C_{11}H_{19}N_2O_2^+$), page S-43 (calcd for $C_{29}H_{39}N_2O_2^{<+>}$), page S-44: (calcd for $C_{17}H_{31}N_2O_2^{<+>}$), and page S-52 (calcd for $C_{37}H_{33}F_6N_2O_2^{<+>}$).

(e) Page S-6 (line 1) and elsewhere: For the purity of solvents, the term "anhydrous" instead of "dry" would be better.

(f) Page S-20 (line 2) and elsewhere, e.g. page S-21 (bottom): The term "diastereo mixture" should be "mixture of diastereomers". In addition, the whole statement "Although compound **S9d** is a diastereo mixture, both 1H and ^{13}C NMR spectra were observed as a single isomer" is mistaken. It should read: "Although compound **S9d** should be a mixture of diastereomers, both 1H and ^{13}C NMR spectra do not reflect this but the presence

of a single isomer". - Similar correction is needed elsewhere.

(g) Page S-21 (line 1) and elsewhere: The term "white solid" should be "colorless solid".

(h) Page S-46: As mentioned above, the isomer ratio of **2q** and **2q'** (as per yield) does not agree with the ratio given in Figure 2 of the main text.

(i) Pages S-56 and S-57: The energy difference calculated by DFT for isomers **9a** and **9a'** and, respectively, for isomers **9b**, **9b'** and **9b''** are in line with those calculated for homocyclic Fenestranes having the all-*cis*-configuration, on the one hand, and the *cis,cis,cis,trans*- or even *cis,trans,cis,trans*-configurations, on the other.

Reviewer #3 (Remarks to the Author):

The manuscript entitled "One-step syntheses of diaza-dioxa-fenestranes via the sequential [3+2]cycloadditions of linear precursors and their structural analyses" reports the synthesis and characterization of diaza-dioxa-fenestranes via the sequential [3+2] cycloaddition reactions. The manuscript is interesting, but requires consideration of some important points as given below:

1. The authors report that replacing the carbon atoms at the non-bridgehead positions in the fenestrane skeleton with nitrogen and oxygen atoms results in planarization of the central quaternary carbon and represents the first example of planarization. This is a significant aspect of the study defining the novelty. The authors have performed DFT calculations to confirm the stereochemistry only. In my opinion, DFT calculations should be performed to analyze the stability of the structures to establish from conformational analysis that the planar structure is the most energetically feasible one. Also a section should be added in the manuscript for the theoretical calculations to analyze the experimental outcome.

2. 1,3-dipole should be replaced by the new terminology "Three atom component (TAC)" throughout the manuscript as the electronic structure of several three atom components deviates from the conventional dipolar electronic structure. Please refer to *Molecules* 21:1319; *Eur J Org Chem* 2019:267-282 for the terminology.

3. The influence of solvent on the reaction rates obtained experimentally should be calculated energetically through DFT calculations to establish the fruitful interplay of theory and experiment.

In this manuscript, Fuse, S. et al. describe the synthesis of aza-oxa-fenestranes starting from simple allenic substrates through formation of di-nitrones or di-nitrile oxides, reacting in two sequential (3+2)-cycloaddition reactions. The bis-cycloaddition reactions are performed in simple operating conditions and the process is attractive to generate aza-oxo-fenestranes. Starting from bis-aldehydic substrates, bis-nitrones are generated in situ by condensation of the bis-aldehydes with hydroxylamines in basic conditions and the subsequent bis-cycloaddition is performed at 110°C in TCE. Complementarily, the same bis-aldehydes can be transformed into their corresponding bis-oximes, which are then oxidized into the corresponding bis-nitrile oxides, which bis-cycloadd to generate the corresponding fenestranes. These compounds have been derivatized through coupling reactions (on the benzylic groups present on the N atoms), reductions of isoxazolidine (obtained from nitrones) or functionalization of the isooxazoline (from nitrile oxides).

The reaction is interesting and the oxo-aza-fenestranes thereby generated are attractive species to be examined. The idea of having an internal allene as electron-rich reactive bis-dipolarophile is also appealing.

However, the manuscript does not discuss into sufficient details the bis-cycloaddition processes. For instance, relative reactivity of the different substrates, diastereoselectivity of the first cycloaddition taking place (endo/exo with nitrones), diastereoselectivity of the second (selectivity relative to the first cycle, endo/exo with nitrones) should be much more detailed (by additional computations for instance and the results should be discussed. It seems that when a non-substituted link is used, only one diastereomer is generated. Why is it the case? Computations should also help in getting more insight into the reaction mechanism, the more precise role of the central sp hybridized allenic carbon atom, the order of reactivity (big discrepancy in reaction yields is observed, is it linked to reactivity, to stability of the cycloadducts..., the diastereoselectivity ..., by localizing, comparing and analyzing the corresponding TSs, which should not be too difficult considering the structures.

When substrates with branched links are considered, mixtures of diastereomers are generated (compounds 2n-2q and 5c). Giving more details about how this can occur, why such a mixture is generated and how this could be improved would be important.

The manuscript mentions that the aza-oxo structure is interesting for the planarization induced by the heteroatoms. More details about how the authors think this takes place and how this can be extended to other structures would broaden the scope of these results. Here again, a more detailed analysis of the structures (orbitals, electron densities...) would help.

In conclusion, these bis-(3+2) processes are attractive and generate compounds of interest. However, much more details about the structures and reaction mechanisms are required to understand and possibly extend these results. In the present form, according to this referee, they are too scarce to be published in *Nature Communications*.

Reviewer #1 (Remarks to the Author):

In this manuscript, Fuse, S. et al. describe the synthesis of aza-oxa-fenestranes starting from simple allenic substrates through formation of di-nitrones or di-nitrile oxides, reacting in two sequential (3+2)-cycloaddition reactions. The bis-cycloaddition reactions are performed in simple operating conditions and the process is attractive to generate aza-oxo-fenestranes. Starting from bis-aldehydic substrates, bis-nitrones are generated in situ by condensation of the bis-aldehydes with hydroxylamines in basic conditions and the subsequent bis-cycloaddition is performed at 110°C in TCE. Complementarily, the same bis-aldehydes can be transformed into their corresponding bisoximes, which are then oxidized into the corresponding bis-nitrile oxides, which bis-cycloadd to generate the corresponding fenestranes. These compounds have been derivatized through coupling reactions (on the benzylic groups present on the N atoms), reductions of isoxazolidine (obtained from nitrones) or functionalization of the isooxazoline (from nitrile oxides). The reaction is interesting and the oxo-aza-fenestranes thereby generated are attractive species to be examined. The idea of having an internal allene as electron-rich reactive bis-dipolarophile is also appealing.

These comments are appreciated.

However, the manuscript does not discuss into sufficient details the bis-cycloaddition processes.

For instance, relative reactivity of the different substrates,

A significant difference in yields was observed in the synthesis of diaza-dioxa-fenestranes **2**, as the reviewer pointed out. However, we could not analyze in detail the reactivity of the employed substrates in the bis-cycloaddition processes. Because of the unstable cyclization precursors, nitrone-containing allenes **3** were in situ generated from the corresponding aldehyde and used for the bis-cycloaddition processes without isolation. Therefore, it was difficult to accurately estimate the reactivity of the cyclization precursors in the bis-cycloaddition.

On the other hand, a similar level of yields (66%–72%) was observed in the synthesis of double-bond-containing [5.5.5.5]diaza-dioxa-fenestranes **5a-5c** from the corresponding nitrile oxides. Therefore, we did not discuss the influence of the substrate structure on the yields.

The developed approach afforded the diaza-dioxa-fenestrane **2r** containing a six-membered ring in a good yield (65%), whereas the synthesis of **2s** and **2t** containing two 6- or 7-membered rings resulted in a low yield (12%) or no detection of the product, respectively. Because the yields of in situ preparation of these nitrone-containing cyclization precursors for the synthesis of **2r-2t** should be similar owing to the similarity of the substrate structure, the observed difference in the yields of **2r-2t** is attributed to the difference in their ring strains. In the case of the synthesis of double-bond-containing [5.5.5.5]diaza-dioxa-fenestrane **5**, the product **5d** containing a six-membered ring was obtained in a moderate yield (36%), whereas the synthesis of **5e** containing two 6-membered rings resulted in no detection of the product. Therefore, the bis-cyclization of **4** appeared to be more significantly affected by the ring strain than the bis-cyclization of **3**. We added a discussion of the relative reactivity of substrates **3** and **6** in the synthesis of **2r-2t** and **5d-5e** to the manuscript.

Page 6, 1st paragraph, line 3 from the bottom

We revised the text as follows.

Before revision:

[5,5,5,6]Diaza-dioxa-fenestrane **2r** and [5.6.5.6]diaza-dioxa-fenestrane **2s** were obtained in yields of 65 and 12%, respectively, although [5.7.5.7]diaza-dioxa-fenestrane **2t** was not obtained. These results demonstrate the robustness of the developed approach.

After revision:

[5,5,5,6]Diaza-dioxa-fenestrane **2r** and [5.6.5.6]diaza-dioxa-fenestrane **2s** were obtained in yields of 65 and 12%, respectively, while [5.7.5.7]diaza-dioxa-fenestrane **2t** was not obtained, probably because of its higher ring strain.

Page 6, 2nd paragraph, line 8

We added the following text.

Although the one-step sequential cycloaddition chemistry involving nitrones **3** allowed the synthesis of the diaza-dioxa-fenestrans containing up to two six-membered rings, the developed chemistry involving nitrile oxides **6** allowed the synthesis of the diaza-dioxa-fenestrans containing up to one six-membered ring. The latter chemistry appeared to be more significantly affected by the ring strain. Using the developed approach, fenestrans with different ring sizes were constructed through sequential cycloaddition for the first time.

diastereoselectivity of the first cycloaddition taking place (endo/exo with nitrones), diastereoselectivity of the second (selectivity relative to the first cycle, endo/exo with nitrones) should be much more detailed (by additional computations for instance and the results should be discussed. It seems that when a non-substituted link is used, only one diastereomer is generated. Why is it the case? Computations should also help in getting more insight into the reaction mechanism, the more precise role of the central sp hybridized allenic carbon atom, the order of reactivity (big discrepancy in reaction yields is observed, is it linked to reactivity, to stability of the cycloadducts..., the diastereoselectivity ..., by localizing, comparing and analyzing the corresponding TSs, which should not be too difficult considering the structures.

We greatly appreciate this suggestion. We performed extensive DFT calculations for the sequential [3+2]cycloadditions over the past 5 months and added a discussion of the reaction mechanism to the manuscript, as follows:

Pages 10-12

We only obtained all-*cis*-fused diastereomers (*c,c,c,c*-[5.5.5.5]fenestrane containing isoxazolidine rings **2** and

c,c-[5.5.5]fenestrane containing isoxazoline rings **5**) in our study. No generation of *trans*-fused fenestranes was observed in the NMR analysis of crude products, for the syntheses of both **2** and **5**. DFT calculation at the B3LYP³⁴/6-31G+(d,p)³⁵⁻³⁸ level of theory was performed to elucidate the mechanism of the sequential [3+2]cycloadditions on the basis of the report of Nguyen et al.³⁹ (Fig. 4). Alkyl substituents were replaced with methyl groups to reduce the calculation cost.

We performed a DFT calculation of the sequential cycloaddition affording **2**. The calculation results for the first [3+2]cycloaddition of the nitron precursors with the *E*-configuration (**SM_E**) and *Z*-configuration (**SM_Z**) affording **IM_{cis}** (intermediate for the all-*cis*-fused diastereomer of the fenestrane) and **IM_{trans}** (intermediate for the *trans*-fused diastereomer of the fenestrane) are shown in Fig. 4a. The comparison of four pathways (*exo* and *endo* cyclizations of **SM_E** and **SM_Z** substrates) suggested that the *exo* cyclization of **SM_E** via the transition state **TS_{E,exo}** is the most energetically favored pathway affording **IM_{cis}**. Meanwhile, the suggested most energetically favored pathway affording **IM_{trans}** is *endo* cyclization of **SM_E** via the transition state **TS_{E,endo}**. The calculated activation energy difference is 1.5 kcal/mol, and the **TS_{E,exo}** leading to **IM_{cis}** is energetically favored over **TS_{E,endo}** leading to **IM_{trans}**. As expected, the two orthogonal p-orbitals of the allene appear to facilitate the approach of reaction sites in **TS_{E,exo}**. **IM_{cis}** appears to have a similar energy level to **IM_{trans}**.

The calculation results for the second [3+2]cycloaddition of the nitron intermediates with the *E*-configuration (**IM_E**) and *Z*-configuration (**IM_Z**) affording **TM_{cis}** (experimentally obtained all-*cis*-fused diastereomer) and **TM_{trans}** (experimentally not obtained *trans*-fused diastereomer) are shown in Fig. 4b. The comparison of four pathways (*exo* and *endo* cyclizations of **IM_E** and **IM_Z** intermediates) again suggests that the *exo* cyclization of **IM_E** via the transition state **TS_{E,exo}** is the most energetically favored pathway affording **TM_{cis}**. The suggested most energetically favored pathway affording **TM_{trans}** is again *endo* cyclization of **IM_E** via the transition state **TS_{E,endo}**. The calculated activation energy difference is 7.2 kcal/mol, and the **TS_{E,exo}** affording **TM_{cis}** is energetically favored over **TS_{E,endo}** affording **TM_{trans}**. In addition, **TM_{cis}** is significantly more stable than **TM_{trans}**. These calculation results explain why only the all-*cis* diastereomer **TM_{cis}** was experimentally obtained.

We could not calculate transition states affording a *trans*-fused diastereomer in the case of [3+2]cycloaddition of the nitrile oxide precursor, because of the extremely strained structure. The calculated pathway affording the all-*cis*-fused diastereomer of the fenestrane **5a** is presented in Table S10 in the Supporting Information.

Fig. 4 DFT calculation for [3+2]cycloadditions. **a**, DFT calculation for the first [3+2]cycloaddition. SM_E^a : nitrone substrate with the *E*-configuration, SM_Z^a : nitrone substrate with the *Z*-configuration, $TS_{E,exo}$: transition state of *exo* cyclization of SM_E , $TS_{E,endo}$: transition state of *endo* cyclization of SM_E , $TS_{Z,exo}$: transition state of *exo* cyclization of SM_Z , $TS_{Z,endo}$: transition state of *endo* cyclization of SM_Z . IM_{cis} : bicyclic intermediate for the all-*cis*-fused diastereomer of the fenestrane. IM_{trans} : bicyclic intermediate for the *trans*-fused diastereomer of the fenestrane. **b**, DFT calculation for the second [3+2]cycloaddition. IM_E^a : nitrone intermediate with the *E*-configuration, IM_Z^a : nitrone intermediate with the *Z*-configuration, $TS_{E,exo}$: transition state of *exo* cyclization of IM_E , $TS_{E,endo}$: transition state of *endo* cyclization of IM_E , $TS_{Z,exo}$: transition state of *exo* cyclization of IM_Z , $TS_{Z,endo}$: transition state of *endo* cyclization of IM_Z . TM_{trans} : *trans*-fused diastereomer of the fenestrane. TM_{cis} : all-*cis*-fused diastereomer of the fenestrane. ^aNeither the energy barrier for tautomerizations between SM_E and SM_Z nor IM_E and IM_Z were calculated.

When substrates with branched links are considered, mixtures of diastereomers are generated (compounds 2n-2q and 5c). Giving more details about how this can occur, why such a mixture is generated and how this could be improved would be important.

We attempted to explain the observed moderate diastereoselectivities in the synthesis of 2n-2q, 5b, and 5c using DFT calculations. However, the calculated energy difference was small. This is not surprising, because the observed diastereomeric ratio was approximately 1:1–1:3. For these reasons, we did not discuss the

diastereoselectivities in the manuscript.

The manuscript mentions that the aza-oxo structure is interesting for the planarization induced by the heteroatoms. More details about how the authors think this takes place and how this can be extended to other structures would broaden the scope of these results. Here again, a more detailed analysis of the structures (orbitals, electron densities...) would help.

Reviewer #2 (comment 8) pointed out that the flattening effect of the aza-oxo structure is weak and possibly due to the packing effect. We consider our determined angles of **2a** to be reliable because the DFT calculation results were consistent with the X-ray results. However, the difference of the angles between **2a** and **10** was insignificant, as Reviewer #2 pointed out. Therefore, we replaced the original expression for the flattening effect of the aza-oxo structure with a weaker expression and did not discuss the flattening effect in detail.

In conclusion, these bis-(3+2) processes are attractive and generate compounds of interest. However, much more details about the structures and reaction mechanisms are required to understand and possibly extend these results. In the present form, according to this referee, they are too scarce to be published in Nature Communications.

We performed the DFT calculations and added the results to the manuscript, in accordance with the reviewer's comments. We appreciate this reviewer and believe that the manuscript was significantly improved from the original version and is now suitable for publication in *Nature Communications*.

Reviewer #2 (Remarks to the Author):

The authors report in great detail and in a remarkably broad scope on the synthesis of heterocyclic fenestranes of unprecedented constitution. The work presented in this manuscript is sound, of high scientific and technical quality. Also, it has a very high level of originality and deals with a particularly fascinating topic of organic chemistry.

These comments are appreciated.

The molecular skeleton of fenestranes is characterized by a tetracyclic core bearing a central atom shared by all of the four rings. The fenestrane skeleton can be regarded as comprising two crosswise fused spirane units or, alternatively, two laterally fused bicyclo[x.y.0]alkane units. The stereochemistry of fusion of the four rings can vary, but most of the known fenestranes have all-cis-fused configuration (c,c,c,c), which is energetically most favorable because of by far the least steric strain. In a rather small subset of fenestranes that are known by experiment, one or even two of the four peripheral bridgeheads are unsaturated, which considerably increases the strain. However, fenestrane chemistry has gained most attention because of the flattening effect

of the quadruple ring fusion on the configuration of the central carbon atom. Although true “planarization” of a tetracoordinate C-atom has never been achieved experimentally in an organic compound, the potential to construct novel fenestranes with increasing degree of flattening continues to be scientifically tempting and challenging. Based on these considerations, the present work is highly encouraging for at least three reasons.

(1) The synthesis strategy is novel and particularly impressing, since the tetracyclic framework of the title compounds is constructed in a single step from linear precursors. In other words, none of the four rings of the target fenestrane needs to be preformed and stepwise ring assembly can be avoided. To the best of my knowledge, this approach has never been pursued before. This important progress has become reality now by employing linear allenes that bear two in-situ-generated 1,3-dipolar functionalities at their α - and ω -positions. Hence, two sequential [3 + 2]-cycloaddition reactions take place in a single synthesis set and with high efficiency. The allene-based bis-dipoles were generated from suitable allene- α,ω -dialdehydes or allene- α,ω -bis-aldoximes in, admittedly, quite elaborate stepwise syntheses.

These comments are appreciated.

(2) The simplicity of this crucial step allows a large chemical variability in accessing novel N,O-heterocyclic fenestranes. Specifically, two N-O bonds form the rims of one of the two spirocyclic subunits which renders framework chiral. One single but also more substituents can be easily incorporated owing to the fact that the central allene unit of the fenestrane precursors is constructed in a nonsymmetric manner. The use of two related but different strategies, namely, use of nitrones, $R-CH=N+(O^-)-R'$, and nitrile oxides, $R-C\equiv N^+-O^-$, as 1,3-dipolar cycloaddition components, allows the construction of both N-saturated and N-unsaturated fenestranes. In the latter case, two C-N double bonds can be generated with ease at the opposite bridgehead positions of a strained cis,cis-[5.5.5.5]fenestradiene skeleton. Further, these double bonds have been used successfully to introduce two allyl groups at the unsaturated bridgeheads. Another remarkable success – which may be overlooked easily in the given context – is the successful synthesis of a (carbocyclic!) spiro[4.4]nonane bearing one singly-bonded functional (amino and hydroxy) group at each of the four α -positions of its neopentane core (viz., compound 8 from fenestrane 2a, Figure 2). Please see remark (9) below.

These comments are appreciated.

(3) The results reported in this work enables further insights into the effect of the fenestrane periphery on the flattening of configuration of the central carbon atom. As mentioned above, incorporation of heteroatoms into the fenestrane framework represents a valuable extension of this theme. Here, the elucidation of the bond angles at the central carbon atom of compounds 2a, 2r and 5a represents an important progress. In particular, this hold true for fenestradiene 5a containing two unsaturated bridgeheads. The central bonds angles found by single crystal X-ray analysis (and DFT calculations) of this structure ($\alpha = 134.7^\circ$ and $\beta = 114.9^\circ$) are

comparably large and of similar size as those of a [4.5.5.5]fenestrene derivative reported by Keese et al. Moreover, the present shows that ring-size effects in the new diazadioxafenestranes are similar to those known from previous work in the carbocyclic series.

These comments are appreciated.

On the basis of all these considerations, I am convinced that the results presented in the manuscript add tremendously to the knowledge of fenestrene chemistry and provide inspiring implications to other, more general fields of chemistry.

These comments are appreciated.

Nevertheless, I found a number of aspects that may deserve correction or improvement prior to acceptance for publication. These points are listed below. In particular, please note the important critique given in (8).

* * * * *

(1) Some parts of the text are somewhat inelegant and deserve improvement. Other minor items to be corrected are listed here as well.

(i) Page 1 (Abstract) and elsewhere (e.g. page 5 and page 9, Discussion): The term “acyclic allenyl precursors with orthogonal p-orbitals” should be shortened because the reader will know that allene have two mutually orthogonal π -orbitals.

The revised manuscript has been edited by a professional English-language editing service (Editage, Tokyo, Japan) again.

The following text was revised.

Abstract

Before revision:

nonbranched acyclic allenyl precursors with orthogonal p-orbitals

After revision:

nonbranched acyclic allenyl precursors

Page 5, line 4 from the bottom

Before revision:

using two types of 1,3-dipole-containing allene with orthogonal p-orbitals: nitron **3a** and nitrile oxide **6a**.

After revision:

using two types of allenes **3a** and **6a** containing nitron and nitrile oxide, respectively.

Discussion

Before revision:

nonbranched acyclic allenyl precursors with orthogonal p-orbitals

After revision:

nonbranched acyclic allenyl precursors

– (ii) *Page 1 (Abstract) and elsewhere: The word “planarization” should be used with care since the configuration of the central C-atom of fenestranes has remained very far from being planar. The term “flattening” would be more appropriate in most cases.*

Abstract, Pages 1, 2, 3, 8, 9, and 12

The word “planarization” was revised to “flattening.”

– (iii) *Page 1 (Abstract) and elsewhere (e.g. page 8, line 5): The chemical name “c,c,c,c-[5.5.5.5]isoxazolines” is incorrect (and meaningless). What is actually meant is here is “cis,cis,cis,cis-[5.5.5.5]festrans containing two isooxazoline units”. Please do not mix different terms of nomenclature.*

The following text was revised.

Abstract

c,c,c,c-[5.5.5.5]isoxazolines -> c,c-[5.5.5.5]fenestranes containing two isooxazoline rings

Page 9, lines 3–5

c,c,c,c-[5.5.5.5]isoxazolidine **2a** -> c,c,c,c-[5.5.5.5]fenestrane containing isoxazolidine rings **2a**

c,c,c,c-[5.5.5.6]isoxazolidine **2r** -> c,c,c,c-[5.5.5.6]fenestrane containing isoxazolidine rings **2r**

c,c,c,c-[5.5.5.5]isoxazoline **5a** -> c,c-[5.5.5.5]fenestrane containing isoxazoline rings **5a**

Page 9, lines 10 and 11

[5.5.5.5]isoxazolidine **2a** -> [5.5.5.5]diazadioxafenestrane **2a**

[5.5.5.6]isoxazolidine **2r** -> [5.5.5.6] diazadioxafenestrane **2r**

Page 9, lines 13 and 14

[5.5.5.5]isoxazolidine **2a** -> [5.5.5.5]fenestrane containing isoxazolidine rings **2a**

[5.5.5.5]isoxazoline **5a** -> [5.5.5.5]fenestrane containing isoxazoline rings **5a**

Fig. 3, legend and ref. 33

c,c,c,c-[5.5.5.5]isoxazolidine **2a** -> c,c,c,c-[5.5.5.5]fenestrane containing isoxazolidine rings **2a**

c,c,c,c-[5.5.5.6]isoxazolidine **2r** -> *c,c,c,c*-[5.5.5.6]fenestrane containing isoxazolidine rings **2r**
c,c,c,c-[5.5.5.5]isoxazoline **5a**. -> *c,c*-[5.5.5.5]fenestrane containing isoxazoline rings **5a**

Page 12, line 15

c,c-[5.5.5.5]isoxazoline **5a** -> *c,c*-[5.5.5.5]fenestrane containing isoxazoline rings **5a**

– (iv) Page 2, line 2: The phrase “... formation of multiple bonds in single steps ...” is incorrect and misleading.

We revised the text as follows.

Page 2, line 2,

Before revision:

formation of multiple bonds in single steps are very useful in organic synthesis

After revision:

formation of multiple bonds in a single step are very useful in organic synthesis

– (v) Page 2, 2nd paragraph: As to precursor **B** in Figure 1, the authors mention one single ring but the structure contains a second (dashed) one. This should be clarified in the text. Improvement along these lines is also necessary in the legend to Figure 1.

We revised the text as follows.

Page 2, 3rd paragraph, lines 3 and 4

Before revision:

The first approach involves the use of precursor **B**, which contains a ring present in the tetracyclic fenestrane structure (Fig. 1, **a**).

After revision:

The first approach involves the use of precursor **B**, which contains one or two rings present in the tetracyclic fenestrane structure (Fig. 1, **a**).

Fig. 1, legend

Before revision:

Sequential cycloaddition of precursor **B** containing a ring found in the fenestrane tetracycle (the first approach).

After revision:

Sequential cycloaddition of precursor **B** containing one or two rings found in the fenestrane tetracycle (the

first approach).

– (vi) Page 6, line 6 (and maybe elsewhere, e.g. page 7, legend to Figure 2): The term “arylbromide” for the para-bromophenyl residue in compound **2g** is unfortunate.

Page 6, lines 5–7

Before revision:

Good yields (68–71%) were obtained using arylbromide **2g**, O-Si-bond-containing **2h**, and substrates **2i** and **2j** containing acid-labile THP, and furyl groups, respectively.

after revision

Good yields (68–71%) were obtained in the syntheses of **2g** containing Ph-Br bonds, **2h** containing O-Si bonds, and **2i** and **2j** containing acid-labile THP and furyl groups, respectively.

Fig. 2, legend

Before revision:

Derivatizing aryl bromide **2g**

After revision:

Derivatizing **2g**

– (vii) Page 6, 2nd-last line of 1st paragraph: The word “although” should be replaced by “while” or “whereas”.

The word “although” was replaced with “while.”

– (viii) Page 7, line 2: The phrase “... unique cyclic compounds” appears to be too vague an expression here (maybe better: “complex heteropolycyclic?”).

The phrase “unique cyclic compounds” was revised to “complex heterocyclic compounds.”

– (ix) Pages 7-8: Just as an example, the terminology should be checked thoroughly throughout. Writing “... as the tetrahedral carbon flattens ...” is really not good.

Page 8, line 4 from the bottom

The phrase “these angles increase as the tetrahedral carbon flattens” was revised to “these angles increase as the quaternary carbon center flattens.”

We rechecked the entire manuscript, and the phrases “central quaternary carbon” and “quaternary carbon atom” were replaced with “quaternary carbon center.”

(2) Figure 1: The novel strategy is displayed in part d of this figure but not sufficiently clearly. While the dashed bonds in E and F are obviously meant to comprise both the nitron and nitrile oxide variants, and their meaning in F can be understood, the presence of the two residues “R” in E excludes the intended perception of nitrile oxide groups. In my view, the two variants could easily be presented in two separate lines here. Also, this would favorably highlight these two variants early in the paper.

This comment is appreciated. Two lines for sequential [3+2][3+2]cycloadditions via the nitron precursor and nitrile oxide precursor are shown in Fig. 1 (d and e).

(3) Page 5 (top): It appears that a comment on entry 8 (no base) is missing.

Page 5, line 11

The following sentence was added to the manuscript.

“The desired **5a** was also obtained in an acceptable yield in the absence of the base (entry 8, 62%).”

(4) Page 5 (top): The phrase “... the higher stability of the less-acidic bisoxime 4a ...” is misleading. I assume that the authors intend to address the lower C-H acidity of the bis-oximes? Otherwise, they should exert some O-H acidity.

Page 5, line 10,

The misleading text “which is possibly ascribable to the higher stability of the less-acidic bisoxime **4a** under basic conditions” was removed from the manuscript.

(5) The scheme at the top of Table 2 displays the nitrile oxide groups in a nonlinear form. In my view, linear presentation ($R-C \equiv N^+-O^-$) should be preferred.

The structure of **6a** in the figure for Table 2 was revised in accordance with the reviewer’s suggestion.

(6) Page 6, end of 1st paragraph and/or possibly also end of 2nd paragraph: A comment on the (limited) ring-size tolerance of the synthesis methodology should be included here.

In accordance with the reviewer’s comment and the comment from Reviewer #1, we revised the text as follows.

Page 6, 1st paragraph, lines 1–3 from the bottom

Before revision:

[5,5,5,6]Diaza-dioxa-fenestrane **2r** and [5.6.5.6]diaza-dioxa-fenestrane **2s** were obtained in yields of 65 and

12%, respectively, although [5.7.5.7]diazadioxafenestrane **2t** was not obtained.

After revision:

[5.5.5.6]Diazadioxafenestrane **2r** and [5.6.5.6]diazadioxafenestrane **2s** were obtained in yields of 65 and 12%, respectively, while [5.7.5.7]diazadioxafenestrane **2t** was not obtained, probably because of its higher ring strain.

Page 6, 2nd paragraph, line 5 from the bottom

The following text was added to the manuscript.

Although the one-step sequential cycloaddition chemistry involving nitrones **3** allowed the synthesis of the diazadioxafenestrans containing up to two six-membered rings, the developed chemistry involving nitrile oxides **6** allowed the synthesis of the diazadioxafenestrans containing up to one six-membered ring. The latter chemistry appeared to be more significantly affected by the ring strain. Using the developed approach, fenestrans with different ring sizes were constructed through sequential cycloaddition for the first time.

(7) Page 7, Figure 2, and Supporting Information (Experimental): While most of the data presented here are in line with those given in the SI, there seems to be a mistake in the diastereomer ration of compound 2q. The ratio of the "up"- and "down" isomers is given as 1.4 : 1 in Figure 2 but in the Experimental, the ratio 2q : 2q' = 23 : 33 ($\approx 1 : 1.4$) is given. Please check.

This comment is greatly appreciated. The ratio shown in Fig. 2 is incorrect. The incorrect ratio "up/down = 1.4:1" was revised to the correct ratio "up/down = 1:1.4."

(8) Page 8, 2nd paragraph: I do not completely agree with the discussion on the flattening effects on the conformation of the central C-atom observed by X-ray diffraction (and supported by the DFT calculations). It should be improved. Taken strictly, there are three points: (i) The statement "A comparison of the quaternary-carbon angles in fenestrane 10 with those in diazadioxafenestrane 2a (Fig. 3, a and d) reveals that replacing the carbon atoms at the non-bridgehead positions in the fenestrane skeleton with nitrogen and oxygen atoms results in planarization of the quaternary carbon, and represents the first example of planarization through the introduction of heteroatoms." is incorrect or, at least, exaggerated. In fact, the flattening effect in 2a is very similar to that found in the carbocyclic fenestrane 10. The slight increase of the angles found for 2a may be real but, even then, it may be due to packing effects. Therefore, I recommend choosing a more cautious statement.

The DFT calculation results were consistent with the X-ray results. Therefore, we consider the determined angles of **2a** to be reliable; however, the difference between the angles of **2a** and **10** was not significant, as the reviewer pointed out. We revised the Abstract, Page 8, and Discussion as follows in accordance with the reviewer's suggestion.

Abstract, line 8

Before revision:

which suggested that replacing the carbon atoms at the non-bridgehead positions in the fenestrane skeleton with nitrogen and oxygen atoms results in flattening of the quaternary carbon center, and represents the first example of flattening through the introduction of heteroatoms.

After revision:

which suggested that replacing the carbon atoms at the non-bridgehead positions in the fenestrane skeleton with nitrogen and oxygen atoms results in slight flattening of the quaternary carbon center.

Page 3, 2nd paragraph, line 3 from the bottom

Before revision:

played a significant role

After revision

played a role

Page 9, 2nd paragraph, line 5

Before revision:

A comparison of the quaternary-carbon angles in fenestrane **10** with those in diaza-dioxa-fenestrane **2a** (Fig. 3, **a** and **d**) reveals that replacing the carbon atoms at the non-bridgehead positions in the fenestrane skeleton with nitrogen and oxygen atoms results in flattening of the quaternary carbon, and represents the first example of flattening through the introduction of heteroatoms.

After revision:

A comparison of the quaternary-carbon angles in fenestrane **10** with those in diaza-dioxa-fenestrane **2a** (Fig. 3, **a** and **e**) reveals that replacing the carbon atoms at the non-bridgehead positions in the fenestrane skeleton with nitrogen and oxygen atoms results in slight flattening of the quaternary carbon center.

Page 12, line 10

Before revision:

Our results show that replacing the carbon atoms at the non-bridgehead positions in the fenestrane skeleton with nitrogen and oxygen atoms flattens the quaternary carbon, which represents first example of flattening through the introduction of heteroatoms.

After revision:

Our results indicate that replacing the carbon atoms at the non-bridgehead positions in the fenestrane skeleton with nitrogen and oxygen atoms slightly flattens the quaternary carbon center.

– (ii) *The next statement; “A comparison of the angles in [5.5.5.5]isoxazolidine 2a with those in [5.5.5.6]isoxazolidine 2r (Fig. 3, e) reveals that ring contraction planarizes the quaternary carbon, which is consistent with the previously reported tendency.5” is misleading. Probably, the authors want to state that contracting the ring size from six to five (i.e., 2r → 2a) increases the flattening. However, five-membered rings in fenestranes may be considered a kind of standard; therefore, the statement should be reversed to express simply that incorporation of a six-membered ring in 2r decreases the flattening.*

We revised the text as follows in accordance with the reviewer’s suggestions.

Page 9, 2nd paragraph, line 8

Before revision:

A comparison of the angles in [5.5.5.5]diazadioxafenestrane **2a** with those in [5.5.5.6]diazadioxafenestrane **2r** (Fig. 3, e) reveals that ring contraction flattens the quaternary carbon, which is consistent with the previously reported tendency.

After revision:

A comparison of the angles in [5.5.5.5]diazadioxafenestrane **2a** with those in [5.5.5.6]diazadioxafenestrane **2r** (Fig. 3, e and f) reveals that ring expansion reduces the degree of flattening of the quaternary carbon, which is consistent with the previously reported tendency.

Page 12, line 12

Before revision:

In addition, we experimentally confirmed that both ring contraction and the introduction of double bonds at the bridgehead positions of a fenestrane results in flattening.

After revision:

In addition, we experimentally confirmed that ring expansion reduces the degree of flattening, whereas the introduction of double bonds at the bridgehead positions of a fenestrane increases the degree of flattening.

– (iii) *The discussion on the “flattest” quaternary carbon atom is somewhat unsatisfying. Not only Keese’s fenestrane 11 (central bond angles $\alpha = 134.9^\circ$ and $\beta = 119.2^\circ$) should be considered but also Agosta’s [4.4.4.5]fenestrane ($\alpha = 129.2^\circ$ and $\beta = 128.3^\circ$, *J. Am. Chem. Soc.* 1985, 107, 5732). To avoid, in the same time, the slightly unclear statement “... hence, 5a contains the flattest quaternary carbon atom among previously synthesized heteroatom-containing fenestrane versions”, I suggest to replace the last sentence of this paragraph by: “However, the observed angles in 5a (Fig. 3, f, $\alpha = 134.7^\circ$, $\beta = 114.9^\circ$) are very large and comparable to those of fenestrane 11, which is among the flattest fenestranes known (Fig. 3, b). Compound 5a*

contains the flattest quaternary carbon atom among heteroatom-containing fenestranes known so far.” Of course, displaying Agosta’s [4.4.4.5]fenestrane in Figure 3 and including this detail into the discussion would be even better.

We presented the chemical structure of Agosta’s [4,4,4,5]fenestrane with the angles ($\alpha = 129.2^\circ$ and $\beta = 128.3^\circ$) in Fig. 3c and cited the manuscript (*J. Am. Chem. Soc.* 1985, 107, 5732) as ref. 32. The following text was revised in accordance with the reviewer’s suggestions.

Page 8, line 1 from the bottom

Before revision:

The flattest reported fenestrane **11** has significantly larger angles (Fig. 3, **b**, $a = 134.9^\circ$, $b = 119.2^\circ$)³¹ than those (109.5°) of a non-distorted quaternary carbon (Fig. 3, **c**).

After revision:

The flattest reported fenestrane **11** and **12** have significantly larger angles (Fig. 3, **b**, $a = 134.9^\circ$, $b = 119.2^\circ$;³¹ Fig. 3, **c**, $a = 129.2^\circ$, $b = 128.3^\circ$ ³²) than non-distorted quaternary carbon (109.5° , Fig. 3, **d**).

Page 9, 2nd paragraph, line 4 from the bottom

Before revision:

However, the observed angles in **5a** (Fig. 3, **f**, $a = 134.7^\circ$, $b = 114.9^\circ$) are very large and comparable to those of fenestrane **11**, which is known to be the flattest fenestrane (Fig. 3, **b**); hence, **5a** contains the flattest quaternary carbon among previously synthesized heteroatom-containing fenestrane versions.

After revision:

However, the observed angles in **5a** (Fig. 3, **g**, $\alpha = 134.7^\circ$, $\beta = 114.9^\circ$) are very large and comparable to those of fenestrane **11** and **12**, which are among the flattest fenestranes known (Fig. 3, **b** and **c**). Compound **5a** contains the flattest quaternary carbon center among heteroatom-containing fenestranes discovered thus far.

(9) Page 9, Discussion section: The "Discussion" presented here does not really discuss but, rather, conclude the results but may be acceptable. In any case, however, the “unique spirocycle 8” does deserve some discussions, as stated above.

We added a DFT calculation section (pages 10–12) to discuss the reaction mechanism of the developed sequential cycloaddition. In addition, we revised the following text.

Page 7, line 8 from the bottom

Before revision:

reductive cleavage of the N–O bond in isoxazolidine **2a** afforded spirobicycle **8** in excellent yield (Fig. 2, **d**, 93%).

After revision:

reductive cleavage of the N–O bond in isoxazolidine **2a** afforded spirobicycle **8** in excellent yield (Fig. 2, **d**, 93%). Spiro[4.4]nonane **8**, which was densely functionalized by two amino groups and two hydroxy groups at the neopentyl positions, was obtained as a single diastereomer.

Page 12, line 5

Before revision:

In addition, **2a**, **2g**, and **5a** were further structurally modified to afford more-functionalized derivatives **7a-7d**, and **9a-9b**, as well as the unique spirobicycle **8**.

After revision:

In addition, **2a**, **2g**, and **5a** were further structurally modified to afford more-functionalized derivatives **7a-7d**, and **9a-9b**. Spiro[4.4]nonane **8**, which was densely functionalized by two amino groups and two hydroxy groups at the neopentyl positions, was obtained as a single diastereomer from reductive cleavage of N-O bonds of **2a**.

(10) A further remark: Compound 9b appears to be a promising candidate for a ring-closing metathesis reaction between the two syn-oriented allyl groups. This would produce an interesting, novel centropentacyclic structure.

We appreciate the valuable suggestion from the reviewer. We plan to attempt the reaction in the future and report the results in another manuscript.

(11) And one more - at the very end: The authors give a useful introduction on the chemistry of alicyclic fenestranes and general aspects of flattening. Unfortunately, they do not mention at all that benzannulated fenestranes have also been studied in great detail and also with respect to flattening of the geometry of the central carbon atom. This aspect may be considered.

As the reviewer pointed out, various benzannulated fenestranes have been synthesized and analyzed. Benzene rings are typically fused at the non-bridgehead positions. We observed significant flattening of the quaternary carbon center with the introduction of double bonds at the bridgehead positions. We compared our results with those of the most similar alicyclic fenestranes rather than with those of less similar benzannulated fenestranes.

(12) Supporting Information: The supporting information is very well-presented and provides complete information on the impressive experiments and measurements. However, some details may deserve improvement of correction, still:

(a) Although the synthesis steps are given completely, a general overview in a scheme at the beginning of the

SI (page S-2) would be helpful.

An overview of the syntheses was added to pages S-3 to S-6, in accordance with the reviewer's suggestion.

(b) Page S-3 (top): Numerous incorrect statements appear in the experimental procedures, such as: "After being stirred at room temperature for 2 h, the reaction was quenched ...". Referring the stirring to the reaction is not correct and not good style, given the very frequent cases of this sort throughout.

We checked the entire Supporting Information, and the phrase "After being stirred" was revised to "After the resultant mixture was stirred."

(c) Page S-4 and throughout the experimental procedures: There is no doubt that the calculated accurate mass values of the $[M + Na]^+$ ions measured in most cases are all correct. However, the statement in the form "HRMS (ESI): calcd. for $C_{13}H_{22}O_2$ ($[M+Na]^+$): 233.1512, found: 233.1514" is not correct because the calculated and measured values refer to the sodiated molecular adduct ions. Therefore, it should better read: "HRMS (ESI) of $[M + Na]^+$: calcd for $C_{13}H_{22}O_2Na^+$ 233.1512, found 233.1514" here, and the like elsewhere.

This comment is appreciated. We revised the HRMS data as follows (the style of all the HRMS data was revised).

HRMS (ESI): calcd. for ($[C_{13}H_{22}O_2+Na]^+$): 233.1512, found: 233.1514.

(d) Pages S-32: Similarly, the accurate mass values of the $[M + H]^+$ ions refer to the formula $C_{11}H_{19}N_2O_2^+$, not $C_{11}H_{18}N_2O_2^+$. Therefore, it should better read here: "HRMS (ESI) of $[M + H]^+$: calcd for $C_{11}H_{19}N_2O_2^+$: 441.2384, found 441.2382". - Similar corrections are needed on Page S-36 (calcd for $C_{11}H_{19}N_2O_2^+$), page S-43 (calcd for $C_{29}H_{39}N_2O_2^{<+>}$), page S-44: (calcd for $C_{17}H_{31}N_2O_2^+$), and page S-52 (calcd for $C_{37}H_{33}F_6N_2O_2^{+>}$).

This comment is also appreciated. The style of the HRMS data was revised as follows.

HRMS (ESI): calcd. for ($[C_{11}H_{18}N_2O_2+H]^+$): 441.2384, found: 441.2382.

(e) Page S-6 (line 1) and elsewhere: For the purity of solvents, the term "anhydrous" instead of "dry" would be better.

The word "dry" was revised to "anhydrous" throughout the Supporting Information.

(f) Page S-20 (line 2) and elsewhere, e.g. page S-21 (bottom): The term "diastereo mixture" should be "mixture of diastereomers". In addition, the whole statement "Although compound S9d is a diastereo mixture, both 1H and ^{13}C NMR spectra were observed as a single isomer" is mistaken. It should read:

“Although compound S9d should be a mixture of diastereomers, both 1H and 13C NMR spectra do not reflect this but the presence of a single isomer”. - Similar correction is needed elsewhere.

The text in the Supporting Information was revised as follows, in accordance with the reviewer’s suggestion.

Before revision:

Although compound **XX** is a diastereo mixture, both ¹H and ¹³C NMR spectra were observed as a single isomer.

After revision:

Although compound **XX** should be a mixture of diastereomers, both ¹H and ¹³C NMR spectra do not reflect this but the presence of a single isomer.

(g) Page S-21 (line 1) and elsewhere: The term “white solid” should be “colorless solid”.

The phrase “white solid” was revised to “colorless solid” throughout the Supporting Information.

(h) Page S-46: As mentioned above, the isomer ratio of 2q and 2q’ (as per yield) does not agree with the ratio given in Figure 2 of the main text.

As described previously, the ratio shown in Fig. 2 was incorrect. It was revised.

(i) Pages S-56 and S-57: The energy difference calculated by DFT for isomers 9a and 9a’ and, respectively, for isomers 9b, 9b’ and 9b’’ are in line with those calculated for homocyclic Fenestranes having the all-cis-configuration, on the one hand, and the cis,cis,cis,trans- or even cis,trans,cis,trans-configurations, on the other.

In accordance with the reviewer’s comments, we estimated the relative stability of stereoisomers of **9a** and **9b** via DFT calculations, as shown below. Unsurprisingly, the calculation results suggested that the all-cis diastereomers **9a** and **9b** are significantly more stable than all the other diastereomers **9a’–9a’’** and **9b’–9b’’**. These results were added to the Supporting Information.

Reviewer #3 (Remarks to the Author):

The manuscript entitled "One-step syntheses of diaza-dioxa-fenestranes via the sequential [3+2]cycloadditions of linear precursors and their structural analyses" reports the synthesis and characterization of diaza-dioxa-fenestranes via the sequential [3+2] cycloaddition reactions. The manuscript is interesting,

These comments are appreciated.

but requires consideration of some important points as given below:

1. The authors report that replacing the carbon atoms at the non-bridgehead positions in the fenestrane skeleton with nitrogen and oxygen atoms results in planarization of the central quaternary carbon and represents the first example of planarization. This is a significant aspect of the study defining the novelty. The authors have performed DFT calculations to confirm the stereochemistry only. In my opinion, DFT calculations should be performed to analyze the stability of the structures to establish from conformational analysis that the planar structure is the most energetically feasible one. Also a section should be added in the manuscript for the theoretical calculations to analyze the experimental outcome.

A conformation search was performed for diaza-dioxa-fenestranes **2b** (benzyl groups in **2a** were replaced with methyl groups to reduce the calculation cost) and **5a** as shown below, in accordance with the reviewer's suggestion. The four most stable conformers (1–4) of **2b** as well as the two most stable conformers (1 and 2) of **5a** are presented with the relative energy levels and α and β values in Table S9 in the revised Supporting Information. The calculated α and β values for the most stable conformer 1 of **2b** ($\alpha = 117.6^\circ$; $\beta = 116.8^\circ$) were similar to those for **2a** determined via X-ray crystallography ($\alpha = 117.4^\circ$; $\beta = 117.0^\circ$). Our calculation results indicated that the less stable conformers 2–4 have more flattened quaternary carbon centers. No stable conformer with a less flattened quaternary carbon center than conformer 1 was found in the conformation search. In the case of **5a**, only two stable conformers (1 and 2) were found, presumably owing to its rigid

structure. These conformers are structurally very similar. The results are briefly explained as follows.

Page 9, last paragraph

We performed a conformation search for fenestranes **2b** and **5a**. Details are presented in Table S9 of the Supporting Information. Fenestrane **2b** with methyl groups was used instead of **2a** with benzyl groups to reduce the calculation cost. The calculation results indicated that the experimentally observed conformer of **5a** ($\alpha = 134.7^\circ$, $\beta = 114.9^\circ$) is the most stable. Although conformers with more flattened quaternary carbon centers were found in the conformation search, they are less stable. However, in the case of **5a**, only two conformers, which were structurally similar and had almost identical α and β values, were found. These results indicate that **5a** has a very rigid structure.

Table S9. Conformation search for fenestranes **2b** and **5a**.

compound	conformer	relative energy (kcal/mol) ^a	α	β
2b	1	0.00	117.6°	116.8°
	2	5.23	118.9°	116.8°
	3	5.31	117.6°	118.4°
	4	5.98	118.2°	118.6°
5a	1	0.00	133.1°	117.1°
	2	0.00	133.0°	117.1°

^aB3LYP/6-31G+(d,p)

2. 1,3-dipole should be replaced by the new terminology "Three atom component (TAC) throughout the manuscript as the electronic structure of several three atom components deviates from the conventional dipolar electronic structure. Please refer to *Molecules* 21:1319; *Eur J Org Chem* 2019:267-282 for the terminology.

We used the phrase "1,3-dipole" only twice in the manuscript, and the deletion of the phrase does not have a significant effect on the text. Therefore, we deleted the phrase, as follows.

Page 3, 2nd paragraph, line 5

Before revision:

allenes **E** and **G** containing 1,3-dipoles (nitrones or nitrile oxides) as precursors

After revision:

allene precursors **E** and **G** containing nitrones and nitrile oxides, respectively

Page 5, 1st paragraph, line 4 from the bottom

Before revision:

using two types of 1,3-dipole-containing allene: nitron **3a** and nitrile oxide **6a**

After revision:

using two types of allenes **3a** and **6a** containing nitron and nitrile oxide, respectively

3. The influence of solvent on the reaction rates obtained experimentally should be calculated energetically through DFT calculations to establish the fruitful interplay of theory and experiment.

The energy levels of all the substrates, intermediates, target molecules, and transition states of DFT calculations for sequential [3+2]cycloaddition affording **2** and **5** in vacuum and solvents were compared, as shown below. No obvious differences were observed. Therefore, the influence of the solvent appeared to be insignificant in the examined reactions. These results are presented in Tables S10 and S11 in the revised Supporting Information.

Table S10. DFT calculations for sequential [3+2]cycloaddition of bisnitrile oxide in vacuum and solvents (DCE, THF, and toluene). **SM**: bisnitrile oxide substrate, **TS1**: transition state of the first cyclization of **SM**, **IM**: intermediate for the all-*cis*-fused diastereomer of the fenestrane, **TS2**: transition state of the second cyclization of **IM**, **TM**: all-*cis*-fused diastereomer of the fenestrane **5a**.

entry	solvent	Method	SM	TS1	IM	TS2	TM
1	-	-	0.0	18.2	-39.8 (0.0)	-18.5 (21.3)	-64.9 (-25.2)
2	DCE	PCM	0.0	18.5	-40.5 (0.0)	-17.6 (22.9)	-64.3 (-23.7)
3	DCE	SMD	0.0	19.0	-40.9 (0.0)	-17.4 (23.6)	-64.5 (-23.6)
4	THF	PCM	0.0	18.5	-40.5 (0.0)	-17.7 (22.8)	-64.3 (-23.8)
5	THF	SMD	0.0	19.0	-40.7 (0.0)	-17.2 (23.5)	-64.3 (-23.6)
6	Toluene	PCM	0.0	18.4	-40.1 (0.0)	-17.9 (22.2)	-64.5 (-24.4)
7	Toluene	SMD	0.0	19.0	-39.8 (0.0)	-16.6 (23.2)	-63.2 (-23.4)

Table S11. DFT calculations for sequential [3+2]cycloaddition of nitron in vacuum and solvents (TCE, THF, and toluene). **a**, DFT calculation for the first [3+2]cycloaddition. **SM_E**: nitron substrate with *E*-configuration, **SM_Z**: nitron substrate with *Z*-configuration, **TS_{E,exo}**: transition state of *exo* cyclization of **SM_E**, **TS_{E,endo}**: transition state of *endo* cyclization of **SM_E**, **TS_{Z,exo}**: transition state of *exo* cyclization of **SM_Z**, **TS_{Z,endo}**: transition state of *endo* cyclization of **SM_Z**. **IM_{cis}**: bicycle intermediate for the all-*cis*-fused diastereomer of the fenestrane. **IM_{trans}**: bicycle intermediate for the *trans*-fused diastereomer of the fenestrane. **b**, DFT calculation for the second [3+2]cycloaddition. **IM_E**: nitron intermediate with *E*-configuration, **IM_Z**: nitron intermediate with *Z*-configuration, **TS_{E,exo}**: transition state of *exo* cyclization of **IM_E**, **TS_{E,endo}**: transition state of *endo* cyclization of **IM_E**, **TS_{Z,exo}**: transition state of *exo* cyclization of **IM_Z**, **TS_{Z,endo}**: transition state of *endo* cyclization of **IM_Z**. **TM_{trans}**: *trans*-fused diastereomer of the fenestrane. **TM_{cis}**: all-*cis*-fused diastereomer of the fenestrane.

entry	solvent	Method	SM _E	TS _{E,endo}	IM _{trans}
1	-	-	4.8	23.4	-19.7
2	TCE	PCM	3.6	24.8	-16.8
3	TCE	SMD	3.5	25.4	-16.5
4	THF	PCM	4.1	24.2	-18.1
5	THF	SMD	4.0	24.4	-18.3
6	Toluene	PCM	4.1	24.2	-18.0
7	Toluene	SMD	4.0	25.0	-17.6

entry	solvent	Method	SM _Z	TS _{Z,endo}	IM _{cis}
1	-	-	0.0	28.5	-19.9
2	TCE	PCM	0.0	29.8	-17.0
3	TCE	SMD	0.0	30.2	-16.6
4	THF	PCM	0.0	29.2	-18.3
5	THF	SMD	0.0	29.2	-18.5
6	Toluene	PCM	0.0	29.3	-18.2
7	Toluene	SMD	0.0	29.8	-17.7

entry	solvent	Method	SM _E	TS _{E,exo}	IM _{cis}
1	-	-	4.8	21.9	-19.9
2	TCE	PCM	3.6	23.4	-17.0
3	TCE	SMD	3.5	23.9	-16.6
4	THF	PCM	4.1	22.7	-18.3
5	THF	SMD	4.0	22.9	-18.5
6	Toluene	PCM	4.1	22.7	-18.2
7	Toluene	SMD	4.0	23.4	-17.7

entry	solvent	Method	SM _Z	TS _{Z,exo}	IM _{trans}
1	-	-	0.0	25.9	-19.7
2	TCE	PCM	0.0	27.1	-16.8
3	TCE	SMD	0.0	27.6	-16.5
4	THF	PCM	0.0	26.6	-18.1
5	THF	SMD	0.0	26.8	-18.3
6	Toluene	PCM	0.0	26.7	-18.0
7	Toluene	SMD	0.0	27.3	-17.6

entry	solvent	Method	IM _E	TS _{E,endo}	TM _{trans}
1	-	-	4.3	29.0	-7.5
2	TCE	PCM	3.8	29.8	-4.4
3	TCE	SMD	3.9	30.5	-3.6
4	THF	PCM	4.0	29.4	-5.9
5	THF	SMD	4.1	30.0	-5.6
6	Toluene	PCM	3.9	29.4	-5.7
7	Toluene	SMD	4.1	30.6	-4.7

entry	solvent	Method	IM _Z	TS _{Z,exo}	IM _{trans}
1	-	-	0.0	51.8	-7.5
2	TCE	PCM	0.0	53.1	-4.4
3	TCE	SMD	0.0	54.1	-3.6
4	THF	PCM	0.0	52.5	-5.9
5	THF	SMD	0.0	53.2	-5.6
6	Toluene	PCM	0.0	52.6	-5.7
7	Toluene	SMD	0.0	53.9	-4.7

entry	solvent	Method	IM _E	TS _{E,exo}	TM _{cis}
1	-	-	4.3	21.8	-24.2
2	TCE	PCM	3.8	23.3	-21.1
3	TCE	SMD	3.9	24.2	-20.4
4	THF	PCM	4.0	22.6	-22.6
5	THF	SMD	4.1	23.1	-22.4
6	Toluene	PCM	3.9	22.7	-22.5
7	Toluene	SMD	4.1	23.8	-21.5

entry	solvent	Method	IM _Z	TS _{Z,endo}	IM _{cis}
1	-	-	0.0	22.5	-24.2
2	TCE	PCM	0.0	24.0	-21.1
3	TCE	SMD	0.0	25.0	-20.4
4	THF	PCM	0.0	23.3	-22.6
5	THF	SMD	0.0	23.9	-22.4
6	Toluene	PCM	0.0	23.4	-22.5
7	Toluene	SMD	0.0	24.5	-21.5

Other minor revisions

Revision of compound name in the manuscript

“*c,c,c,c*-[5.5.5.5]isoxazoline **5a**” was revised to “*c,c*-[5.5.5.5]isoxazoline **5a**” throughout the manuscript and Supporting Information.

Page 4, Results, 1st paragraph, line 4

Before revision:

entries, 1–3, for details, see Table S1 in the Supporting Information for details

After revision:

entries 1–3; see Table S1 in the Supporting Information for details

Ref. 36

Before revision:

Hehre, W. J. & Ditchfield, R.; Pople, J. A. self

After revision:

Hehre, W. J., Ditchfield, R. & Pople, J. A. Self

Ref. 38

Before revision:

Clark, T., Chandrasekhar, J., Spitznagel, G. W.; Schleyer, P. v. R.

After revision:

Clark, T., Chandrasekhar, J., Spitznagel, G. W. & Schleyer, P. v. R.

A data availability statement was added to the manuscript, as follows.

Data Availability

The authors declare that the data for this study are available within the manuscript and its Supplementary Information files and are also available from the corresponding author. The nuclear magnetic resonance (NMR) spectra, experimental procedures, characterization results for all products, and DFT calculations are presented in the Supplementary Information file. The X-ray crystallographic coordinates for the structures reported in this article have been deposited at the Cambridge Crystallographic Data Centre (CCDC), under deposition numbers CCDC-2233428 (for **2a**), CCDC-2233429 (for **2r**), and CCDC-2233425 (for **5a**). These data can be obtained free of charge from the Cambridge Crystallographic Data Centre via www.ccdc.cam.ac.uk/structures.

Acknowledgement

We added the following text.

“Pd/C was generously provided by Kawaken Fine Chemicals, Japan.”

REVIEWER COMMENTS

Reviewer #1 (Remarks to the Author):

In this revised version of their paper, Fuse, S. et al. have greatly improved their manuscript and answered most of the reviewers' comments. The additional DFT calculations indeed help to understand the mechanism of the bis-cycloaddition process taking place. Accordingly, I think that the manuscript is now suitable for publication in Nature Communications, after a minor addition concerning the discussion about the second (3+2) cycloaddition process described in Fig 4b.

The first step leads to IMcis in a model reaction. For the second step, this IMcis is considered as IME and IMZ, with IME being less stable by 4.3 kcal.mol⁻¹. From IME, TSE,exo (at 21.8 kcal.mol⁻¹) is more favourable than TSE,endo and leads to the experimentally observed TMcis. This is perfectly fine.

If considering the more stable IMZ, TSZ,endo (22.5 kcal.mol⁻¹) is the most favourable and also leads to TMcis experimentally observed.

The difference between TSE,exo and TSZ,endo ($\Delta\Delta G^\ddagger = 0.7$ kcal.mol⁻¹), is small, too small to be significant considering DFT calculations and both pathways through these two TSs leading to the same product should be considered and discussed in the manuscript, assuming that the IM Z to E isomerization TS is less energetic than the cycloaddition TSs, which seems reasonable.

Additional minor remark:

The cycloaddition are described as [3+2]cycloaddition (with no space). According to IUPAC nomenclature, 1,3-dipolar cycloadditions are (3+2) cycloadditions (number of atoms) but [4+2] cycloaddition (electrons). The author should consider modifying the "[3+2] cycloaddition" terms into "(3+2) cycloaddition" throughout the manuscript.

Reviewer #2 (Remarks to the Author):

Please see attached report.

Reviewer #3 (Remarks to the Author):

Authors have addressed all the comments and I find the manuscript suitable for publication in the journal.

Reviewer's report on the manuscript

One-step syntheses of diaza-dioxa-fenestranes via the sequential [3+2]cycloadditions of linear precursors and their structural analyses

by Shinichiro Fuse, Hiroki Ishikawa, Hisashi Masui, and Takashi Takahashi

submitted to *Nature Communications* (NCOMMS-23-52237A) – Revised version

My general assessment of the manuscript has not been changed because of the extremely high originality of the work. The authors have responded in detail to my comments and suggestions for improvement, although one critical suggestion has been misunderstood and should be re-considered. I have also checked the presentation and discussion of the computational work based on DFT; part of that work is not presented in an acceptable manner. All this is addressed in the list provided below. In summary, in my view the manuscript requires a second thorough revision before acceptance.

* * * * *

Items of minor importance

- (1) Most of the formal errors have been corrected. However, please check where the terms "isooxazolinidine" and "isooxazoline" should be replaced by "isoxazolinidine" and "isoxazoline" (see line 35 for example).
- (2) Lines 38-40: The text has been revised in a misleading manner. Instead of " Sequential cycloaddition reactions are particularly effective because they facilitate the formation of multiple bonds both regio- and stereoselectively in one step¹⁻³ " it should read " Sequential cycloaddition reactions are particularly effective because they facilitate the multiple formation of bonds both regio- and stereoselectively in one step¹⁻³ ".
- (3) Lines 42-43 and beyond: As mentioned in my report (item 11) on the original version of the manuscript (maybe in a somewhat reluctant way), I would consider it scientifically important and also fair to mention and or at least quote the chemistry of benzo-annulated fenestranes just in the early introductory parts of this manuscript. The authors' response concerns the flattening effect of double bonds or benzene rings; whereas my comment does not touch this at all. Rather, the overview on fenestrane chemistry should simply help the reader to orient him/herself appropriately. Therefore, the authors may reconsider referring to relevant publications, such as that in *Angew. Chem. Int. Ed.* **56**, 12356-12360 (2017) and *Chem. Rev.* **106**, 4885-4925 (2006), which also refer to flattening effects at the central quaternary carbon atom.
- (4) Line 48: "nonplaner" (typo).
- (5) Line 73: The article "a" at the end of the line should be deleted.
- (6) Figure 1 has been improved very well.
- (7) Line 125: "... The desired **5a** ..." is no good style; it should read "... The desired product **5a** ..." , or the like.
- (8) Line 154: "[5,5,5,6]Diaza-dioxa-fenestrane **2r**" should be "[5.5.5.6]Diaza-dioxa-fenestrane **2r**".

(9) Lines 164-170: The newly inserted comments starting here are not well written (logic) and require correction and also improvement (discussion). I suggest that, instead of

“Although the one-step sequential cycloaddition chemistry involving nitrones **3** allowed the synthesis of the diaza-dioxa-fenestranes containing up to two six-membered rings, the developed chemistry involving nitrile oxides **6** allowed the synthesis of the diaza-dioxa-fenestranes containing up to one six-membered ring. The latter chemistry appeared to be more significantly affected by the ring strain. Using the developed approach, fenestranes with different ring sizes were constructed through sequential cycloaddition for the first time.”

The text should maybe read as follows:

“Whereas the one-step sequential cycloaddition chemistry involving nitrones **3** allowed the synthesis of the diaza-dioxa-fenestranes containing up to two six-membered rings, the developed chemistry involving nitrile oxides **6** allowed the synthesis of the diaza-dioxa-fenestranes containing one single six-membered ring only. In the latter cases, ring strain appears to be a much more limiting factor, presumably due to the presence of two double bonds at the bridgeheads of the fenestrane framework. Using the developed approach, fenestranes with different ring sizes were constructed through sequential cycloaddition for the first time.

(10) Lines 173 and 175: Please delete the article “a” in “... in the presence of a transition-metal catalysts ...” and insert the article “the” in “... which afforded desired products 7a–7d ...”.

(11) Figure 2 has been improved very well.

(12) Line 201: Please correct to “... The flattest reported fenestranes 11 and 12 ...” (plural).

(13) Lines 204-206: The statement

“Our synthesized racemic *c,c,c*-[5.5.5.5]fenestrane containing isoxazolidine rings **2a**, *c,c,c,c*-[5.5.5.6]fenestrane containing isoxazolidine rings **2r**, and *c,c*-[5.5.5.5]fenestrane containing isoxazoline rings **5a** were analyzed by X-ray crystallography, ...”

is somewhat “heavy” and should be shorted. For example:

“The racemic [5.5.5.5]- and [5.5.5.6]fenestranes **2a** and **2r**, respectively, containing isoxazolidine rings and the [5.5.5.5]fenestrane **5a** containing isoxazoline rings **5a** were analyzed by X-ray crystallography, ...”.

(14) Line 206: The reference “32” should be shifted to appear after the word “crystallography”.

(15) Line 214: In “... the angles in [5.5.5.5]fenestrane containing isoxazolidine rings **2a** ...”, it should be read “... the angles in [5.5.5.5]fenestrane **2a** containing isoxazolidine rings ...”.

(16) Lines 221-234: The authors have added a presentation and discussion on the results of the DFT calculations. In my view, the part needs in-depth correction and improvement.

(i) The authors do not address the contents of Tables S7 and S8. On reading, it first appeared to me that their mentioning Table S9 was a mistaken but then I realized that Table S9 does in fact contain a search on conformers of compound **2b** and **5a**. However, since the contents of Table S7 and S8 are not addressed at all, this part of the text is extremely misleading. It should be re-written completely. –

Also, the authors state that, reasonable, the *N,N*-dimethyl-substituted fenestrane **2b** was used studied by the calculation, instead of the *N,N*-dibenzyl-substituted fenestrane **2a**. However, all the structural formulas displayed in Table S7 show the dibenzyl analog. This is contradictory! – The authors do not mention at all that they have calculated various ring sizes. – Again, please not the incorrect term “isooxazolidine” instead of “isoxazolidine” in the Table.

(17) Lines 223-224) Moreover, the discussion should be more meaningful: The sentence “The calculation results indicated that the experimentally observed conformer of **5a** ($\alpha = 134.7^\circ$, $\beta = 114.9^\circ$) is the most stable.” is unsatisfying: It is not clear what the authors mean by “conformer”: In Table S8, they display the structures of the two enantiomers of fenestrane **5a** but there is no presentation of different conformers of this compound. Second, they give the values of the two angles α and β at the central carbon atom but, in such a discussion, they should also mention the respective values obtained by the DFT calculation.

(18) Lines 224-227: Further, the sentence “Although the conformers with more flattened quaternary carbon centers were found in the conformation search, they are less stable. On the other hand, structurally very similar, only two conformers with almost consistent α and β values were found in the case of **5a**.”

I am sorry for this extended critique but it the text is really chemically inadequate and far below what would be appropriate in view of the overall quality of the experimental work.

(19) Line 227: In the sentence “These results indicated that the **5a** has a very rigid structure.” The word “compound” should be inserted before “**5a**”.

(20) Figure 3: I recommend strongly to improve the layout of this figure. The text below structure **10** should be rearranged into four lines, such that the first row (**a-d**) could be narrowed.

(21) Line 237: “We only obtained all-*cis*-fused diastereomers ...” should be improved to “We exclusively obtained all-*cis*-fused diastereomers ...” or, probably better, “In this study, all-*cis*-fused [5.5.5.5]fenestrane diastereomers were obtained exclusively.”

(22) Lines 248 and 250: “*exo* cyclization” and “*endo*-cyclization” should have hyphens.

(23) Line 249: I do not understand the meaning of the word “Meanwhile” here.

(24) The amendments suggested for the Supporting Information have all been made in a constructive - and enjoyable - manner.

Response to reviewers' comments

Reviewer #1 (Remarks to the Author):

In this revised version of their paper, Fuse, S. et al. have greatly improved their manuscript and answered most of the reviewers' comments. The additional DFT calculations indeed help to understand the mechanism of the bis-cycloaddition process taking place. Accordingly, I think that the manuscript is now suitable for publication in Nature Communications, after a minor addition concerning the discussion about the second (3+2) cycloaddition process described in Fig 4b. The first step leads to IM_{cis} in a model reaction. For the second step, this IM_{cis} is considered as IM_E and IM_Z, with IM_E being less stable by 4.3 kcal.mol⁻¹. From IM_E, TSE_{,exo} (at 21.8 kcal.mol⁻¹) is more favourable than TSE_{,endo} and leads to the experimentally observed TM_{cis}. This is perfectly fine.

These comments are appreciated.

If considering the more stable IM_Z, TS_{Z,endo} (22.5 kcal.mol⁻¹) is the most favourable and also leads to TM_{cis} experimentally observed. The difference between TSE_{,exo} and TS_{Z,endo} ($\Delta\Delta G^\ddagger = 0.7$ kcal.mol⁻¹), is small, too small to be significant considering DFT calculations and both pathways through these two TSs leading to the same product should be considered and discussed in the manuscript, assuming that the IM_Z to E isomerization TS is less energetic than the cycloaddition TSs, which seems reasonable.

I believe this reviewer's suggestion is quite reasonable. Therefore, we revised the text as follows,

Lines 257-263

Before revision

The comparison of four pathways (*exo* and *endo* cyclizations of **IM_E** and **IM_Z** intermediates) again suggests that the *exo* cyclization of **IM_E** via the transition state **TS_{E,exo}** is the most energetically favored pathway affording **TM_{cis}**. The suggested most energetically favored pathway affording **TM_{trans}** is again *endo* cyclization of **IM_E** via the transition state **TS_{E,endo}**. The calculated activation energy difference is 7.2 kcal/mol, and the **TS_{E,exo}** affording **TM_{cis}** is energetically favored over **TS_{E,endo}** affording **TM_{trans}**. In addition, **TM_{cis}** is significantly more stable than **TM_{trans}**. These calculation results explain why only the all-*cis* diastereomer **TM_{cis}** was experimentally obtained.

After revision

The comparison of four pathways (*exo*- and *endo*-cyclizations of **IM_E** and **IM_Z** intermediates) again suggests that the *exo*-cyclization of **IM_E** via the transition state **TS_{E,exo}** is the most energetically favored pathway affording **TM_{cis}**. Additionally, the *endo*-cyclization of **IM_Z** via the transition state **TS_{Z,endo}** was also suggested as the energetically plausible pathway in the case of second cycloaddition because the calculated activation energy difference between **TS_{E,exo}** and **TS_{Z,endo}** was 0.7 kcal/mol. The most energetically favored pathway affording

TM_{trans} was again suggested to be *endo*-cyclization of IM_E via the transition state $\text{TS}_{E,endo}$. The calculated activation energy difference between $\text{TS}_{E,exo}$ and $\text{TS}_{E,endo}$ was 7.2 kcal/mol, and the $\text{TS}_{E,exo}$ affording TM_{cis} was energetically favored over $\text{TS}_{E,endo}$ affording TM_{trans} . Moreover, TM_{cis} was significantly more stable than TM_{trans} . These results explain the reason for obtaining only the all-*cis* diastereomer TM_{cis} .

Additional minor remark:

The cycloaddition are described as [3+2]cycloaddition (with no space). According to IUPAC nomenclature, 1,3-dipolar cycloadditions are (3+2) cycloadditions (number of atoms) but [4+2] cycloaddition (electrons). The author should consider modifying the “[3+2] cycloaddition” terms into “(3+2) cycloaddition” throughout the manuscript.

All instances of “[3+2]cycloaddition” have been revised to “(3+2) cycloaddition” in our revised manuscript and revised supporting information.

Reviewer #2 (Remarks to the Author):

My general assessment of the manuscript has not been changed because of the extremely high originality of the work. The authors have responded in detail to my comments and suggestions for improvement,

These comments are appreciated.

although one critical suggestion has been misunderstood and should be re-considered. I have also checked the presentation and discussion of the computational work based on DFT; part of that work is not presented in an acceptable manner. All this is addressed in the list provided below. In summary, in my view the manuscript requires a second thorough revision before acceptance.

Items of minor importance

(1) Most of the formal errors have been corrected. However, please check where the terms “isooxazolinidine” and “isooxazoline” should be replaced by “isoxazolinidine” and “isoxazoline” (see line 35 for example).

Line 35

The wrong phrase “isooxazoline” was revised to “isoxazoline.”

Supporting Information

The wrong phrases “isooxazoline” and “isooxazolidine” have been revised to “isoxazoline” and “isoxazolidine,” respectively.

(2) Lines 38-40: The text has been revised in a misleading manner. Instead of “ Sequential cycloaddition reactions are particularly effective because they facilitate the formation of multiple bonds both regio- and stereoselectively in one step1-3 ” it should read “ Sequential cycloaddition reactions are particularly effective

because they facilitate the multiple formation of bonds both regio- and stereoselectively in one step¹⁻³ ”.

Lines 38-39

The text was revised as follows in accordance with this reviewer’s comment,

before revision

Sequential cycloaddition reactions are particularly effective because they facilitate the formation of multiple bonds both regio- and stereoselectively in one step

after revision

Sequential cycloaddition reactions are particularly effective because they facilitate the multiple formation of bonds, both regio- and stereoselectively, in one step

*(3) Lines 42-43 and beyond: As mentioned in my report (item 11) on the original version of the ust manuscript (maybe in a somewhat reluctant way), I would consider it scientifically important and also fair to mention and or at least quote the chemistry of benzo-annulated fenestranes just in the early introductory parts of this manuscript. The authors’ response concerns the flattening effect of double bonds or benzene rings; whereas my comment does not touch this at all. Rather, the overview on fenestrane chemistry should simply help the reader to orient him/herself appropriately. Therefore, the authors may reconsider referring to relevant publications, such as that in *Angew. Chem. Int. Ed.* 56, 12356-12360 (2017) and *Chem. Rev.* 106, 4885-4925 (2006), which also refer to flattening effects at the central quaternary carbon atom.*

I misunderstood the previous comments from this reviewer. I thought the reviewer required the comparison of the flattening effect between the benzene ring and the double bond in the results section, therefore, I did not include the description related to benzene-annulated fenestranes because it seemed difficult to compare the flattening effect directly and precisely. However, now, I have understood the comments. I also think it is fair and reasonable to cite references related to the benzene-annulated fenestranes in the Introduction.

Lines 42-43

References 8 and 9 shown below have been cited in our revised manuscript.

8. Kuck, D. Three-dimensional hydrocarbon cores based on multiply fused cyclopentane and indane units: Centropolyindanes. *Chem. Rev.* **106**, 4885-4925 (2006). [10.1021/cr050546+](https://doi.org/10.1021/cr050546+).
9. Wong, W.-S., Ng, C.-F., Kuck, D. & Chow, H.-K. From fenestrindane towards saddle-shaped nanographenes bearing a tetracoordinate carbon atom. *Angew. Chem. Int. Ed.* **56**, 12356-12360 (2017). [10.1002/anie.201707505](https://doi.org/10.1002/anie.201707505).

(4) Line 48: “nonplaner” (typo).

Line 48

The wrong phrase “nonplaner” was revised to “nonplanar.”

(5) Line 73: The article “a” at the end of the line should be deleted.

Line 75

The unnecessary article “a” in the sentence “using a nonbranched acyclic precursors” was removed.

(6) Figure 1 has been improved very well.

Thank you for your helpful comment in the previous review.

(7) Line 125: “... The desired 5a ...” is no good style; it should read “... The desired product 5a ...”, or the like.

Lines 120 and 125

“Desired **5a**” and “The desired **5a**” were revised to “The desired product **5a**.”

(8) Line 154: “[5,5,5,6]Diaza-dioxa-fenestrane 2r” should be “[5.5.5.6]Diaza-dioxa-fenestrane 2r”.

The wrong phrase “[5,5,5,6]Diaza-dioxa-fenestrane” was revised to “[5.5.5.6]Diaza-dioxa-fenestrane.”

(9) Lines 164-170: The newly inserted comments starting here are not well written (logic) and require correction and also improvement (discussion). I suggest that, instead of

*“Although the one-step sequential cycloaddition chemistry involving nitrones **3** allowed the synthesis of the diaza-dioxa-fenestrans containing up to two six-membered rings, the developed chemistry involving nitrile oxides **6** allowed the synthesis of the diaza-dioxa-fenestrans containing up to one six-membered ring. The latter chemistry appeared to be more significantly affected by the ring strain. Using the developed approach, fenestrans with different ring sizes were constructed through sequential cycloaddition for the first time.”*

The text should maybe read as follows:

*“Whereas the one-step sequential cycloaddition chemistry involving nitrones **3** allowed the synthesis of the diaza-dioxa-fenestrans containing up to two six-membered rings, the developed chemistry involving nitrile oxides **6** allowed the synthesis of the diaza-dioxa-fenestrans containing one single six-membered ring only. In the latter cases, ring strain appears to be a much more limiting factor, presumable due to the presence of two double bonds at the bridgeheads of the fenestrane framework. Using the developed approach, fenestrans with different ring sizes were constructed through sequential cycloaddition for the first time.*

Thank you for your valuable suggestion. We originally believed that the ring strain was the most important factor in the developed one-step sequential cycloaddition chemistry. However, both calculation results in Tables

S7 and S8 indicate that the ring expansion decreases the degree of flattening. Therefore, I am currently unsure whether the fenestranes containing 6- and 7-membered rings are really more strained than the fenestranes with only 5-membered ring.

I revised the text as follows,

Lines 55-56

before revision

while [5.7.5.7]diazadioxafenestrane **2t** was not obtained, probably because of its higher ring strain.

after revision

while [5.7.5.7]diazadioxafenestrane **2t** was not obtained.

Lines 164-170

before revision

Although the one-step sequential cycloaddition chemistry involving nitrones **3** allowed the synthesis of the diaza-dioxafenestranes containing up to two six-membered rings, the developed chemistry involving nitrile oxides **6** allowed the synthesis of the diaza-dioxafenestranes containing up to one six-membered ring. The latter chemistry appeared to be more significantly affected by the ring strain. Using the developed approach, fenestranes with different ring sizes were constructed through sequential cycloaddition for the first time.

after revision

While one-step sequential cycloaddition chemistry involving nitrones **3** enabled the synthesis of diaza-dioxafenestranes containing up to two six-membered rings, the developed chemistry involving nitrile oxides **6** enabled the synthesis of diaza-dioxafenestranes containing only one six-membered ring. The latter chemistry appeared to be more significantly affected by the ring size. Using the developed approach, fenestranes with different ring sizes were constructed through sequential cycloaddition for the first time.

(10) Lines 173 and 175: Please delete the article “a” in “... in the presence of a transition-metal catalysts ...” and insert the article “the” in “... which afforded desired products 7a–7d ...”.

Lines 173 and 175

I have removed the article “a” and added the article “the.”

(11) Figure 2 has been improved very well.

Thank you for your helpful comment in the previous review.

(12) Line 201: Please correct to “... The flattest reported fenestranes **11** and **12** ...” (plural).

Line 201

The wrong phrase “fenestrane” was revised to “fenestranes.”

(13) Lines 204-206: The statement

“Our synthesized racemic *c,c,c,c*-[5.5.5.5]fenestrane containing isoxazolidine rings **2a**, *c,c,c,c*-[5.5.5.6]fenestrane containing isoxazolidine rings **2r**, and *c,c*-[5.5.5.5]fenestrane containing isoxazoline rings **5a** were analyzed by X-ray crystallography, ...”

is somewhat “heavy” and should be shorted. For example:

“The racemic [5.5.5.5]- and [5.5.5.6]fenestranes **2a** and **2r**, respectively, containing isoxazolidine rings and the [5.5.5.5]fenestrane **5a** containing isoxazoline rings **5a** were analyzed by X-ray crystallography, ...”.

Lines 204-206

Thank you. The text was revised in accordance with this reviewer’s suggestion.

(14) Line 206: The reference “32” should be shifted to appear after the word “crystallography”.

The citation (after revision, the reference number became 35) was moved after the word “crystallography.”

(15) Line 214: In “... the angles in [5.5.5.5]fenestrane containing isoxazolidine rings **2a** ...”, it should be read “... the angles in [5.5.5.5]fenestrane **2a** containing isoxazolidine rings ...”.

Line 214

The position of the compound number “**2a**” was moved.

Lines 215 and 297

The position of the compound number “**5a**” was also moved.

Lines 232-234 and reference 35

The position of compound numbers “**2a**, **2r**, and **5a**” was also moved.

Lines 237 and 238

The position of compound numbers “**2** and **5**” was also moved.

(16) Lines 221-234: The authors have added a presentation and discussion on the results of the DFT calculations. In my view, the part needs in-depth correction and improvement.

(i) The authors do not address the contents of Tables S7 and S8. On reading, it first appeared to me that their mentioning Table S9 was a mistaken but then I realized that Table S9 does in fact contain a search on conformers

of compound 2b and 5a. However, since the contents of Table S7 and S8 are not addressed at all, this part of the text is extremely misleading. It should be re-written completely. –

I sincerely apologize for confusing the reviewer. As the reviewer pointed out, we did not explain the calculation results shown in Tables S7 and S8. We have added paragraphs as follows,

Supporting Information, Table S7,

To estimate the extent of the flattening effect derived from the ring size, α and β values of the calculated most stable conformers of isoxazolidine fenestranes are shown in Table S7. The α and β values of **2a** ($\alpha = 117.3^\circ$; $\beta = 116.6^\circ$) and **2r** ($\alpha = 114.1^\circ$; $\beta = 114.3^\circ$) based on the DFT calculation were similar to those of **2a** ($\alpha = 117.4^\circ$; $\beta = 117.0^\circ$, Fig. 3) and **2r** ($\alpha = 114.4^\circ$; $\beta = 114.2^\circ$, Fig. 3) based on the X-ray crystallographic analysis. The calculation results indicated that the ring contraction increases the degree of flattening of the quaternary carbon center, whereas the ring expansion decreases the degree of flattening. This tendency is consistent with that previously reported.²¹

Supporting Information, Table S8,

To estimate the extent of the flattening effect derived from the ring size, α and β values of the calculated most stable conformers of isoxazoline fenestranes are shown in Table S8. The α and β of **5a** ($\alpha = 133.1^\circ$; $\beta = 117.1^\circ$) based on the DFT calculation were similar to those of **5a** ($\alpha = 134.7^\circ$; $\beta = 114.9^\circ$, Fig. 3) based on the X-ray crystallographic analysis. Again, the calculation results indicated that the ring contraction increases the degree of flattening of the quaternary carbon center, whereas the ring expansion decreases the degree of flattening.

In addition, the compound numbers **2a**, **2r**, **2s**, **5a**, **5d**, and **5e** were added to Tables S7 and Table S8 for clarity.

Also, the authors state that, reasonable, the N,N-dimethyl-substituted fenestrane 2b was used studied by the calculation, instead of the N,N-dibenzyl-substituted fenestrane 2a. However, all the structural formulas displayed in Table S7 show the dibenzyl analog. This is contradictory! – The authors do not mention at all that they have calculated various ring sizes. – Again, please not the incorrect term “isooxazolidine” instead of “isoxazolidine” in the Table.

Although we calculated *N,N*-dimethyl-substituted fenestrane **2b** for Table S9, we calculated *N,N*-dibenzyl-substituted fenestrane **2a** for Table S7. Therefore, chemical structures in Tables S7 and S9 are accurate. We performed the DFT calculation 2 years ago for Table S7 using **2a**. We performed the conformation search for **2b** this year after receiving the reviewers' comments. Then, we intentionally used structurally simple **2b** for Table S9 to avoid overlooking conformations, because **2a** containing rotatable benzyl groups raises calculation cost and the risk of overlooking conformations. As a result, the calculated α and β values for the most stable conformer 1 of **2b** ($\alpha = 117.6^\circ$; $\beta = 116.8^\circ$) were similar to those for **2a** ($\alpha = 117.3^\circ$; $\beta = 116.6^\circ$). Therefore, now, I think both calculations for Tables S7 and S9 are reliable.

The wrong phrases “isooxazolidine” and “isooxazoline” were revised to “isoxazolidine” and “isoxazoline” as previously described.

*(17) Lines 223-224) Moreover, the discussion should be more meaningful: The sentence “The calculation results indicated that the experimentally observed conformer of **5a** ($\alpha = 134.7^\circ$, $\beta = 114.9^\circ$) is the most stable.” is unsatisfying: It is not clear what the authors mean by “conformer”: In Table S8, they display the structures of the two enantiomers of fenestrane **5a** but there is no presentation of different conformers of this compound. Second, they give the values of the two angles α and β at the central carbon atom but, in such a discussion, they should also mention the respective values obtained by the DFT calculation.*

I am very sorry, but we forgot to delete meaningless calculation results for enantiomers of **2a**, **2r**, and **5a** (Enantiomers should have completely consistent energy levels) in Tables S7 and S8 in our original supporting information. These unnecessary results were removed from our revised supporting information. As we did not explain the calculation results shown in Tables S7 and S8, this made our supporting information further confusing (Detailed explanations were added to our revised supporting information as previously described). The calculations shown in Tables S7 and S8 were performed to estimate the extent of the flattening effect derived from the ring size, therefore, only α and β values of “the most stable conformers” were indicated for clarity. Instead, the detailed conformation search results including respective values, α and β , for compounds **2b** and **5a** were shown in Table S9.

I also sincerely apologize for our incorrect statements in lines 221-227. Both the compound number and α and β values were wrong.

Lines 221-227

We revised the text as follows,

before revision

We performed a conformation search for fenestranes **2b** and **5a** (see Table S9 of the Supporting Information for details. Fenestrane **2b** with methyl groups was used instead of **2a** with benzyl groups to reduce the calculation cost). The calculation results indicated that the experimentally observed conformer of **5a** ($\alpha = 134.7^\circ$, $\beta = 114.9^\circ$) is the most stable. Although the conformers with more flattened quaternary carbon centers were found in the conformation search, they are less stable. On the other hand, structurally very similar, only two conformers with almost consistent α and β values were found in the case of **5a**. These results indicated that the **5a** has a very rigid structure.

after revision

We performed a conformation search for fenestranes **2b** and **5a**. To reduce the calculation cost, fenestrane **2b** with methyl groups was used instead of **2a** with benzyl groups. The four most stable conformers 1-4 of **2b** and

the two most stable conformers 1 and 2 of **5a** are shown with relative energy levels and α and β values in Table S9 of the Supporting Information. The chemical structure of **2a** experimentally observed via X-ray crystallographic analysis ($\alpha = 117.4^\circ$, $\beta = 117.0^\circ$) was consistent with the calculated most stable conformer 1 of **2b** ($\alpha = 117.6^\circ$, $\beta = 116.8^\circ$). Although the conformers 2-4 of **2b** with more flattened quaternary carbon centers were found in the conformation search (Table S9), they were less stable. However, only two conformers with almost consistent α and β values and similar structures were found in the case of **5a**. These results indicated that the compound **5a** has a very rigid structure.

I also revised the text as follows to increase clarity,

Line 207

before revision

“which revealed α and β values consistent with those determined by DFT”

after revision

“which revealed α and β values consistent with those of the most stable conformers determined by DFT”

Supporting Information, Table S9,

before revision

The calculated α and β values for the most stable conformer 1 of **2b** ($\alpha = 117.6^\circ$; $\beta = 116.8^\circ$) were similar to those for **2a** determined by X-ray crystallography ($\alpha = 117.4^\circ$; $\beta = 117.0^\circ$).

after revision

The calculated α and β values for the most stable conformer 1 of **2b** ($\alpha = 117.6^\circ$; $\beta = 116.8^\circ$) were similar to those for **2a** determined by DFT calculation ($\alpha = 117.3^\circ$; $\beta = 116.6^\circ$) and X-ray crystallography ($\alpha = 117.4^\circ$; $\beta = 117.0^\circ$).

*(18) Lines 224-227: Further, the sentence “Although the conformers with more flattened quaternary carbon centers were found in the conformation search, they are less stable. On the other hand, structurally very similar, only two conformers with almost consistent α and β values were found in the case of **5a**.”*

I am sorry for this extended critique but it the text is really chemically inadequate and far below what would be appropriate in view of the overall quality of the experimental work.

Again, I sincerely apologize for our insufficient and wrong statement in lines 221-227. The texts were revised as previously described.

*(19) Line 227: In the sentence “These results indicated that the **5a** has a very rigid structure.” The word “compound” should be inserted before “**5a**”.*

Line 227

The phrase “compound” was added before “**5a**.”

*(20) Figure 3: I recommend strongly to improve the layout of this figure. The text below structure **10** should be rearranged into four lines, such that the first row (**a-d**) could be narrowed.*

The layout of Figure 3 was revised in accordance with this reviewer’s comment.

(21) Line 237: “We only obtained all-cis-fused diastereomers ...” should be improved to “We exclusively obtained all-cis-fused diastereomers ...” or, probably better, “In this study, all-cis-fused [5.5.5.5]fenestrane diastereomers were obtained exclusively.

The text was revised as follows,

before revision

We only obtained all-*cis*-fused diastereomers (*c,c,c,c*-[5.5.5.5]fenestrane containing isoxazolidine rings **2** and *c,c*-[5.5.5.5]fenestrane containing isoxazoline rings **5**) in our study.

after revision

In this study, all-*cis*-fused diastereomers (*c,c,c,c*-[5.5.5.5]fenestrane **2** containing isoxazolidine rings and *c,c*-[5.5.5.5]fenestrane **5** containing isoxazoline rings) were obtained exclusively.

(22) Lines 248 and 250: “exo cyclization” and “endo-cyclization” should have hyphens.

All the phrases “*exo* cyclization” and “*endo* cyclization” both in the manuscript and supporting information were revised to “*exo*-cyclization” and “*endo*-cyclization.”

(23) Line 249: I do not understand the meaning of the word “Meanwhile” here.

The phrase “Meanwhile” was revised to “However.”

(24) The amendments suggested for the Supporting Information have all been made in a constructive -and enjoyable - manner.

Thank you for your helpful comment in the previous review.

Reviewer #3 (Remarks to the Author):

Authors have addressed all the comments and I find the manuscript suitable for publication in the journal.

This comment is appreciated.

Other minor revisions

Reference 2, title of the paper,

“Recent Advances” was revised to “Recent advances.”

Reference 12, title of the paper,

“Rhazinilam-Leuconolam-Leuconoxine” was revised to “Rhazinilam-leuconolam-leuconoxine.”

Reference 16, title of the paper,

“Silphinene” was revised to “silphinene”.

Reference 20, title of the paper,

“Synthesis of [4.6.4.6]Fenestradienes and [4.6.4.6]Fenestrenes based on an 8p-6p-Cyclization-Oxidation cascade.” was revised to “Synthesis of [4.6.4.6]fenestradienes and [4.6.4.6]fenestrenes based on an 8 π -6 π -cyclization-oxidation cascade.”

Reference 32, title of the paper,

“[5.5.5.5]Fenestrane” was revised to “[5.5.5.5]fenestrane.”

Reference 33, title of the paper,

“[4.5.5.5]Fenestrane” was revised to “[4.5.5.5]fenestrane.”

REVIEWERS' COMMENTS

Reviewer #1 (Remarks to the Author):

In this new version, the authors have addressed all the comments/recommendations of the referees and I find the manuscript suitable for publication in Nature Communications.